# ATS-YOLOv7: A Real-Time Multi-Scale Object Detection Method for UAV Aerial Images Based on Improved YOLOv7

**Heng Zhang** , **Faming Shao \*** , **Xiaohui He** , **Weijun Chu, Dewei Zhao, Zihan Zhang and Shaohua Bi**

College of Field Engineering, Army Engineering University of PLA, Nanjing 210007, China;
zhangheng4216@sina.com (H.Z.); gcbhxh@aeu.edu.cn (X.H.); 13770533312@139.com (W.C.);
zhaodewei533@126.com (D.Z.); zzh2023@aeu.edu.cn (Z.Z.); 17337434454@163.com (S.B.)
* Correspondence: shaofaming@aeu.edu.cn; Tel.: +86-185-4985-4591

**Abstract:** The objects in UAV aerial images have multiple scales, dense distribution, and occlusion, posing considerable challenges for object detection. In order to address this problem, this paper proposes a real-time multi-scale object detection method based on an improved YOLOv7 model (ATS-YOLOv7) for UAV aerial images. First, this paper introduces a feature pyramid network, AF-FPN, which is composed of an adaptive attention module (AAM) and a feature enhancement module (FEM). AF-FPN reduces the loss of deep feature information due to the reduction of feature channels in the convolution process through the AAM and FEM, strengthens the feature perception ability, and improves the detection speed and accuracy for multi-scale objects. Second, we add a prediction head based on a transformer encoder block on the basis of the three-head structure of YOLOv7, improving the ability of the model to capture global information and feature expression, thus achieving efficient detection of objects with tiny scales and dense occlusion. Moreover, as the location loss function of YOLOv7, CIoU (complete intersection over union), cannot facilitate the regression of the prediction box angle to the ground truth box—resulting in a slow convergence rate during model training—this paper proposes a loss function with angle regression, SIoU (soft intersection over union), in order to accelerate the convergence rate during model training. Finally, a series of comparative experiments are carried out on the DIOR dataset. The results indicate that ATS-YOLOv7 has the best detection accuracy (*mAP* of 87%) and meets the real-time requirements of image processing (detection speed of 94.2 FPS).

**Keywords:** UAV aerial images; object detection; YOLOv7; AF-FPN; transformer encoder; SIoU

## 1. Introduction

At present, the development of UAV technology has reached a certain level. Multi-type and multi-function UAV products have brought great convenience in production and life contexts, as well as providing broader conditions for scientific research in the fields of aerial photography, transportation, patrol inspection, mapping, and so on, through the use of UAVs [1–3]. As an important information resource, UAV aerial image data have been widely used in artificial intelligence [4], agricultural and industrial production [5], urban and environmental monitoring [6], military intelligence reconnaissance [7], and other fields, due to their ease of access, large data scale, rich information, and high value. Therefore, it is very necessary and meaningful to study UAV aerial imagery.

Using object detection technology [8] to study aerial images is one of the most popular research directions at present; however, UAV aerial photography is vulnerable to interference conditions due to various factors such as the shooting equipment, environment, and scene, potentially resulting in blurred images, poor contrast, unclear texture, or dramatic scale changes, thus posing great difficulties for object detection. In order to fully solve this problem, significant research work on intelligent detection technology has been carried out. For example, considering the task of the real-time ground multi-scale object detection of clustered UAVs, one article [9] has improved the basis of the bidirectional parallel

multi-branch feature pyramid network (BPMFPN) and proposed a new detection model called BPMFPN. This model strengthens the expression ability of each scale feature layer in aerial images by constructing a bidirectional parallel pyramid network, then integrates the detection network into a single detector. The primary detector BPMPN-Det was verified on the UAVDT data set, and achieved high precision. X. Zhang et al. [10] proposed a real-time detection model for small objects in UAV imagery based on an improved ASFF-YOLOv5s, in order to address the impact of objects with large scale differences on real-time detection in images captured by UAVs. On the basis of the original YOLOv5, the shallow feature fusion was strengthened and the adaptive spatial feature fusion (ASFF) was improved, so that the model could improve the abilities of multi-scale feature fusion and small object feature extraction. When X Hou et al. [11] studied how to improve the noisy and high-frequency images collected by unmanned helicopters. They first used multi-module anchor-free detectors to balance and optimize the training samples, then proposed a confrontation module with a central weight to solve the problem of weak local features. Their experimental results proved that the proposed methods were efficient on VisDrone2020. At present, most multi-level visual detectors based on deep learning have a high false negative rate when detecting objects in UAV aerial images. B.M. Albaba et al. [12] proposed an integrated network SyNet based on a pretrained CenterNet and Cascade R-CNN, which combines multi-level and single-stage detectors to reduce the above phenomena and enhance the detection performance of single-stage detectors. For the challenging category of tiny targets in aerial images, anchor-based detectors may reduce the quality of tag allocation. C. Xu et al. [13] used a new evaluation method based on the normalized Wasserstein distance (NWD) and a new strategy based on RanKing's allocation (RKA) to detect small objects. The RKA is embedded into the anchor detector to improve the label allocation and provide sufficient training information for the model, such that the detector is more reliable for training and performance verification. Although object detection technology for aerial images has been very successful to date, it is still not accurate for high-precision bearing prediction. For this reason, Q. Ming et al. [14] proposed a detection method based on task interleaving and direction estimation (TIOENet) by combining a variety of strategies such as posterior hierarchical comparison (PHA) tags and balanced alignment loss. The authors used it to solve the misplacement problem in detection progress, the imbalance loss problem in the prediction process, and the prediction accuracy problem of angle deviation.

Many types of aerial image object detection methods were described above. Considering these methods, we can see that the current object detection algorithms are specialized for specific problems and do not have the versatility of multi-field applications. Therefore, professional research on object detection for drone aerial images needs to be strengthened, which would also pave the way for the development of neural networks based on deep learning.

In order to further improve the performance of detection models for complex UAV aerial images, this paper proposes a target detection method for UAV aerial images based on the findings of previous studies. First, we introduce a feature pyramid network, AF-FPN, to solve the problems of multi-scale objects and small objects generated in large-scene UAV aerial photography. AF-FPN is used to reduce the information loss in the feature mapping process and improve the network's ability to represent the feature layer, such that the network can reduce the negative impact caused by the difference of object scales and improve the perception of small objects. Second, in order to address the problem of dense object arrangement, we strengthen the information capture capability of the model by adding a transformer encoder detector head, improving the detection accuracy of the model for such objects. Finally, a new loss function is introduced to improve the original positioning loss in YOLOv7, such that the model can comprehensively consider the angle loss.

Based on the above analysis, the main innovations and contributions of this paper can be summarized as follows:

- A feature pyramid network, AF-FPN, composed of an adaptive attention module (AAM) and a feature enhancement module (FEM), is introduced into the YOLOv7 architecture. On one hand, AF-FPN reduces the information loss caused by the reduction in the number of feature channels during the convolutional mapping process through the AAM. On the other hand, it strengthens the feature representation during the sampling process through the FEM. The introduction of AF-FPN can effectively improve the model's detection accuracy and speed for multi-scale objects.
- This article adds a detection route based on the transformer encoder module on the basis of YOLOv7. The four-head detection structure of this module can effectively alleviate the negative impact caused by a drastic change of object scale, strengthen the feature perception ability of the network for dense and small targets, and greatly improve the detection performance of the network.
- A new loss function, SIoU, is introduced into the algorithm to improve the positioning loss defect of the YOLOv7 loss function, CIoU. SIoU enables the network to fully consider the difference in overlapping area size and angle loss between the ground truth box and the prediction box when regressing. The improved model can accelerate the convergence speed and improve the detection accuracy of occluded objects during training.
- The detection performance of each model on the data set was compared through ablation experiments, SOTA (state-of-the-art) experiments, and real-time comparison experiments, and ATS-YOLOv7 was verified to have a rational combination of modules and efficient detection performance through a balance analysis of precision and speed.

This article is structured as follows: Section 2 mainly discusses the related literature in the field of object detection. Section 3 introduces the overall workflow and specific details of the proposed method. Section 4 details the experimental process, including the dataset selection, comparison experiment, and other work. Section 5 provides the summary and future outlook of this article.

## 2. Related Works

With many years of technological innovation and development, the degree of intelligence of human society has been constantly increasing; as such, the demand for network communication [15], automatic control [16], artificial intelligence, and other technologies in human life has also grown. Corresponding industrial products have rapidly poured into the public eye, such as driverless cars [17], VR human–computer interaction [18], ChatGPT [19], unmanned aerial vehicles, and so on. At present, the most representative AI technology is profoundly changing all aspects of society and is pushing our world into a new era. As an important part of artificial intelligence, computer vision—object detection technology driven by large-scale data—provides important support for various engineering applications. Therefore, it is of great significance to make full use of object detection technology in related scientific research.

Based on previous research on object detection models for UAV aerial images, we aim to make innovative improvements in terms of the following aspects: the YOLO algorithm, aerial object detection, the transformer for object detection, and the loss function. We have learned a lot from previous research, as detailed in the following.

### 2.1. YOLO (You Only Look Once) Algorithm

As early as 1999, W Li et al. in [20], in order to improve the self-organizing mapping neural network model to better adapt to the parallel computing environment, proposed a once learning method (all inputs are learned by the model at one time) to replace the traditional repeated learning method. This kind of human-like learning method improves the efficiency of image processing and lays the foundation for the development of subsequent learning. The YOLO algorithm, presented by Joseph Redmon et al. [21], is a representative one-stage object detection algorithm with the characteristics of high detection accuracy and fast detection speed. It has undergone several version changes over about 10 years,

and there are dozens of official versions and related derivative versions. Among them, the introduction of YOLOv1 [21] in 2015 laid an important foundation for subsequent development of the YOLO series. YOLOv7 [22], as the latest version in the YOLO series, has a more accurate and faster detection performance. It has an updated detection backbone network and prediction structure on the basis of YOLOv5 [23], making it more stable and efficient when facing more challenging detection tasks. Researchers have carried out a lot of research and application work on YOLO series models. For example, YOLO algorithms typically require high-end hardware; however, it is particularly important to carry out real-time detection under the condition of limited computing resources. For this reason, J. Lee et al. [24] discussed the relationship between network cameras and real-time processing related to YOLO detection, and proposed an adaptive-frame-control (AFC) YOLO architecture. This architecture provides high-precision real-time detection for the network by minimizing the total service delay function of the AFC. Using road aerial images to study information such as vehicles, routes, and traffic signs is particularly important for the construction of digital and information-based roads. Z. Li et al. [25] proposed a detection network based on YOLOv5 for tiny targets in road aerial images. They introduced an attention mechanism and the SoftPool module into the architecture to enhance the network's attention to small object feature areas and retain more detailed feature information in the convolution process through pooling operations. In [26], the 6-DOF manipulator (DOF) and the enhanced YOLO algorithm were used to propose a strategy for the safety inspection of personnel at vehicle inspection stations. The authors introduced a dense part into the backbone to achieve a faster detection speed, and improved the network sensitivity by continuously optimizing the anchor box strategy and residual technology. S. Schneidereit et al. [27] designed YOLO models with different architectures for training on complex factory environment data sets in order to embed the YOLO architecture into the Fischertechnik Industry 4.0 application, allowing for monitoring of the manufacturing process in factories. Additionally, a prior shape allocation strategy was integrated to fully improve the model performance.

### 2.2. Aerial Object Detection

Aerial image object detection is one of the most common applications in the field of computer vision. The dead points in the detection process can be due to various reasons, for example: (a) the multi-scale difference of an object causes a pixel change, (b) the distribution of aerial image objects is uneven, (c) the appearance of an object changes due to the light and shooting angle of the image, and (d) dense objects. To solve these problems, Vishnu Chalavadi et al. [28] proposed a multi-scale object detection model, mSODANet, based on hierarchical expansion convolution. They used parallel expansion convolution to explore multi-level features, allowing the network to learn more context information. By introducing a hierarchical expansion network, the model can effectively capture object information and achieves improved detection performance. In [29], the complexity calculation of the general YOLO was improved. A finite convolution layer was introduced into the trunk to expand the receptive field, such that the model can fully learn the characteristics of objects. In the detection part, a discriminant representation is generated for the network through channel and spatial attention modules. The improved model has lightweight and real-time characteristics, and can solve the complex background problem in aerial images. For flood disaster monitoring, remote sensing-based image data are often ineffective, due to the long revisit period and adverse environmental impacts. Therefore, K. Yang et al. [30] studied a depth learning detection model based on YOLOv3 for flood-submerged building images. Through LiDAR (light detection and ranging)-enhanced UAV aerial photography, they collected images with dull light and blurred background, analyzed thermal bridge information, and then detected the submerged key buildings and vegetation. For the detection of aerial images, detection methods based on the anchor frame are popular; however, for images with many small objects, this method often suffers from repeated detection and omission. Therefore, Q. Ming et al. [31] proposed a sparse label allocation strategy (SLA) based on the IoU of anchor crossing, and used the

position-sensitive feature pyramid network (PS-FPN) of the attention module to extract the position features of small objects, allowing them to be accurately located. Considering the problem related to the small volume and occlusion of low-altitude objects, low-altitude detectors based on deep learning cannot effectively extract the context information of small objects. Thus, the authors of [32] created an extended RESNET module (DRM) and feature fusion module (FFM) based on the trigeminal network. With these modules, the model can be applied to low-altitude objects with large-scale changes, and it can well detect the semantic information of low-altitude object features to achieve a higher rise point. A new model for dense small object detection in aerial images is proposed in article [33]. The author combines the backbone of the large selective kernel network (LSKNet) with the DiffusionDet head of the diffusion network, and designs a weighted focus loss function and fine-tuning hyperparameters within the network to optimize the model's regression ability against challenging targets, providing a new benchmark for high-level object detection in aerial images.

### 2.3. Transformer for Object Detection

A transformer was proposed by Vaswani et al. [34] in 2017. It is a neural network based on a self-attention mechanism, which has been widely used in the field of natural language processing (NLP) [35] to deal with sequence-to-sequence speech problems. To date, the transformer has been widely applied to many fields, such as computer vision and speech recognition, due to its efficient parallel information processing ability [36]. In the field of vision, the transformer is often combined with a convolutional neural network (CNN) [37] to handle tasks such as image segmentation, object recognition, and positioning, and the performance of the resulting model often surpasses that of traditional RCNN and CNN models. For typical applications, D. Chen et al. [38] used the decreasing attention gate to improve the transformer, in order to overcome the influence of interference factors of non-critical objects, and used the attention fusion module to allow the network to inherit the attention matrix of the previous layer, thus adding weights to the most critical objects to effectively capture object information. In the field of drone delivery, in order to overcome the impact of limited training sample data, the authors of [39] proposed a network framework based on the hybrid con revolutionary transformer (HCT), which uses a weak supervision mechanism and a small number of sample labels for supervised network learning to better annotate and detect pixel-level images. In [40], a convolution module with a converter was introduced. Global features and a shrink map are extracted through the fusion converter, significantly reducing the amount of calculation in the network. The object query function of the converter is embedded as a set during learning, but the embedding position of each learning set cannot be determined manually and it cannot be optimized centrally. For this reason, Y. Wang et al. [41] designed an object query function based on anchor points. This design method can query by concentrating objects near anchor points or predicting multiple objects in one location, thus addressing the difficulty posed by multiple objects in one area. C. Chen et al. [42] proposed a SwinTD model based on the Swin transformer [43] for detecting foreign objects in tobacco packaging cutting areas. They used the Swin transformer to establish a model of the relationship between foreign objects and the background, while utilizing dense connections to enhance the re-use of local features, thus avoiding over-fitting. In [44], a new application of a transformer in an electronic waste detection method was provided. It provided an efficient image conversion model EWasteNet based on a transformer for accurate classification of electronic waste using dual stream data. The two data streams are used for edge detection and multi-scale feature information capture through Sobel operators and atrous spatial pyramid pooling, respectively. Finally, the feature information is merged and predicted at the decision-making level through a transformer. The experimental results show that this method can effectively analyze the characteristics of electronic waste and improve the accuracy of detecting waste objects.

### 2.4. Loss Function

The loss function is a necessary evaluation indicator and optimization tool in the field of machine learning [45], which is used to measure the performance during algorithm training and detection. It calculates the difference between the results predicted by the model and the true values through various measurement methods, allowing for continuous adjustment of the algorithm parameters to bring the algorithm closer to the true state of the object in the next round of detection. In object detection tasks, reasonable use of different loss functions can allow for the training of better models. For example, the cross-entropy loss function [46] can be used to measure and adjust the classification differences of detection models. Similarly, in regression tasks, the mean squared error loss function [47] is used to evaluate and optimize localization differences. Overall, the loss function is crucial for training efficient and reliable detection models. In this regard, researchers have also carried out a lot of exploration and innovation works. Due to the great contribution of the feedback mechanism to the development of object detection technology, D. Chen et al. [48] developed a control distance IoU (CDIoU, IoU—Intersection Over Union) and related loss function without increasing the model parameter (flop), in order to make up for the shortcomings of the traditional evaluation system and feedback module. In the comparison, this new loss function can effectively feedback the loss of distance to the model. Boundary box regression technology has an impact on the accuracy of object detection. D. Tian et al. [49] introduced a boundary box regression loss function called absolute size (AIoU) to improve the accuracy of object detection. They discussed the limitations of the previous common loss function in the loss of intersection location, then used the AIoU function, which contains a variety of penalty terms, to improve the regression loss of the boundary box. In order to improve the accuracy of object detectors in computational resource-assisted systems (DASs) and maintain efficient detection capabilities in rainy or other harsh environments, Bhaumik Vaidya et al. [50] developed a de-noising network with a custom SSIM loss function in the image detection process. The class weight penalty technique of this network, combined with a trainable color converter, can enhance the detection accuracy and efficiency with respect to small objects. The loss function plays a key role in object detection. In [51], the loss function was optimized from the two aspects of classification and positioning. On one hand, the relationship between the IoU coefficient and classification loss function was established, and the correlation between classification and location was used to reduce the misclassification rate. On the other hand, the gradient inconsistency in DIoU was solved by introducing the Mahalanobis distance between the prediction box and ground truth box.

## 3. Methods

### 3.1. Overview of the Proposed Method

This section describes the overall workflow of the proposed method, ATS-YOLOv7. As shown in Figure 1, the data were collected first. According to the research content, we collected a variety of UAV aerial image datasets (e.g., DIOR [52], DOTA [53], UCAS-AOD [54], and so on) and analyzed the statistics of the data (see Section 4.2 for details). Second, we determined the required data. We conducted a detailed comparative analysis of the collected datasets from five aspects: images, categories, instances, quality, and intra- and inter-class similarities. Then, we determined the datasets required for the model experiment. Finally, object detection was carried out. This section is mainly divided into three steps. The first step is feature extraction. The aerial data were preprocessed at the input of ATS-YOLOv7 and uniformly sized as $640 \times 640$, then sent to the backbone for object feature extraction, from shallow to deep. The second step is multi-scale feature fusion. After the feature layer input from the backbone was reduced by the AAM, the neck FEM and PANet performed the feature enhancement sampling fusion operation. The transformer module improves the ability of the network to capture context information. The third step is object prediction. The four detection heads of ATS-YOLOv7 regressed and

classified the feature layers at different scales from the neck. The internal SIoU function ensures the high precision capability of the network (see Section 3.2 for details).

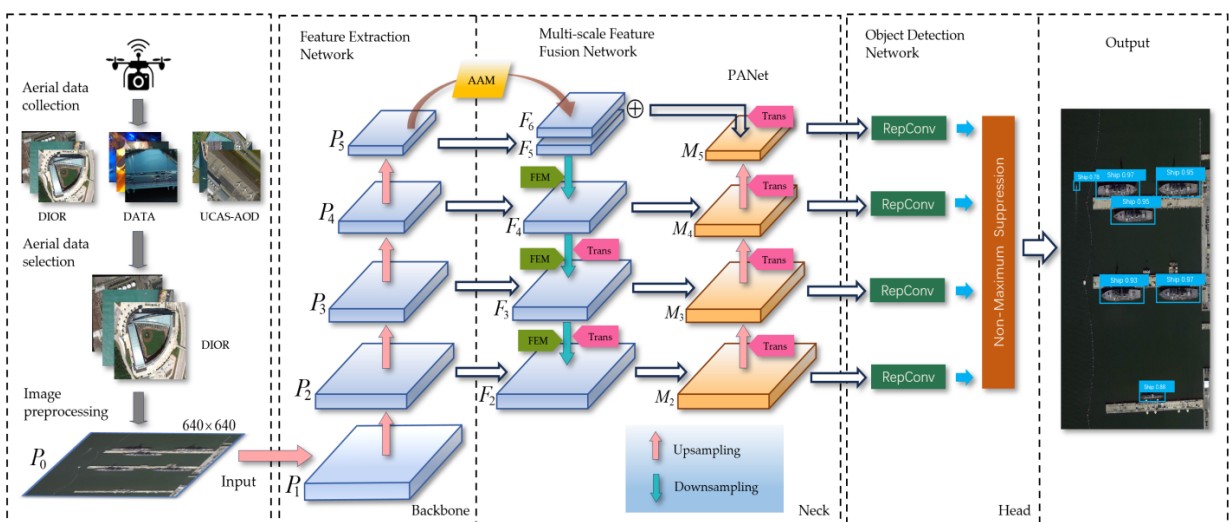

**Figure 1.** The overall workflow of the ATS-YOLOv7 method proposed in this article.

### 3.2. Improved YOLOv7 Network Model (ATS-YOLOv7)

YOLOv7 has a relatively efficient detection capability and inference speed; however, its performance may encounter bottlenecks when detecting complex types of aerial objects such as multi-scale and densely arranged objects. Therefore, this article improves YOLOv7 in three aspects. The specific structure of ATS-YOLOv7 is shown in Figure 2.

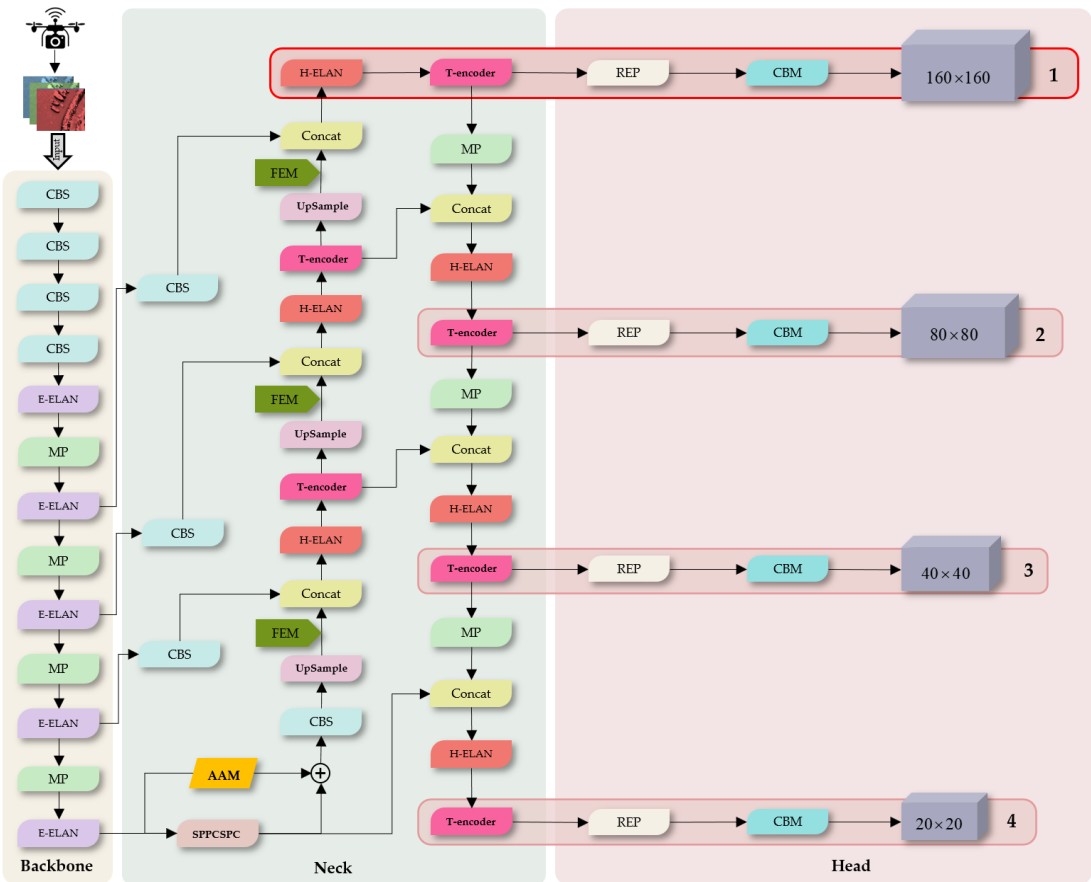

**Figure 2.** Structure and process of ATS-YOLOv7 network.

Specifically, first of all, the maximum feature image size output in YOLOv7 is $80 \times 80$, accounting for one-eighth of the size of the input image; that is, one pixel on the feature image can map to cover eight pixels of the original image. However, this is not sufficient for objects with tiny sizes and few pixels. To this end, we added a transformer encoder block (T-encoder) to the upper image feature processing part of the backbone, and inserted other T-encoder modules into the model neck to form an additional detection route. The resolution of the output feature map of this detection route is one-quarter ($160 \times 160$) of the input map. Due to the high resolution of the feature image on the route and the clear texture and edges of the object, the detection head of this part can detect objects in high-density scenes and objects of small size with high precision. Second, in order to improve the detection accuracy for multi-scale objects, AF-FPN is introduced into the neck. AF-FPN reduces the information loss caused by a reduction of the number of channels in the feature mapping process through the adaptive attention module (AAM) and feature enhancement module (FEM), thus enhancing the representation ability of the feature pyramid and enabling the model to ensure sufficient robustness and efficiency under object size changes. Finally, in order to enhance the positioning and training performance of the network, we improved the anchor frame loss function of the YOLOv7 head. SIoU is introduced, based on the original location loss function CIoU. SIoU allows the model to consider the angle regression loss between the prediction box and the ground truth box by adjusting the difference in the overlapping area between the prediction box and ground truth box with a penalty term. This is beneficial with respect to the convergence speed and detection precision of the model.

The specific steps of each improvement scheme are detailed in the following sections.

### 3.2.1. AF-FPN Block

The existence of multi-scale objects in aerial images affects accuracy and real-time performance in object detection tasks. At present, the classical feature pyramid network is often introduced by researchers in the face of such problems; however, it is difficult to achieve a perfect balance between accuracy and real-time performance in practical applications. Therefore, we introduce an adaptive feature-enhanced pyramid network, AF-FPN [55]. AF-FPN consists of an adaptive attention module (AAM) and a feature enhancement module (FEM). On one hand, AF-FPN reduces the loss of context information in deep features through a reduction of the number of feature channels in the convolution process through the AAM. On the other hand, the FEM is used to improve the feature representation of the feature layer, thus enhancing the calculation speed and real-time performance of the model.

The structure of AF-FPN is shown in Figure 1. In the AF-FPN model, the original image $P_o$ generates the deep feature layer $P_5$ through the backbone network. $P_5$ and the neck feature layer $F_6$ ($F_6$ is $P_5$, which is generated through the path with the AAM) are summed and then fused with features from other paths in a step-by-step manner in the down-sampling process. In the fusion process, the new feature layer expands the receptive field through the FEM to enhance the information perception ability of the algorithm. The PANet part is a sampling path added on the basis of FPN, with the purpose of allowing the network to integrate more levels of features and obtain richer context information.

The structure of the AAM is shown in Figure 3. The working process of the AAM can be divided into two parts. The first part is adaptive pooling. $P_5$ (size $S = H \times W$) obtains the context features at three different scales ($\alpha_1 \times S, \alpha_2 \times S, \alpha_3 \times S$) through the average pooling operation of the adaptive pooling layer. Here, $\alpha$ is the pooling coefficient, with a size in the range of [0.1, 0.5], which changes adaptively with a change in the object size. Then, a $1 \times 1$ convolution is used to make the three context features have the same channel dimension, and a scale feature layer is obtained through bilinear interpolation and up-sampling operations, which is convenient for feature fusion in the following steps. In the second part, feature fusion, the *concat* layer of the spatial attention mechanism merges context feature channels at different scales, and the merged feature layer obtains the

corresponding spatial weight map of the feature layer through the $1 \times 1$ convolution layer, the *ReLU* function activation layer, the $3 \times 3$ convolution layer, and the *sigmoid* function activation layer of the spatial attention mechanism in turn. The spatial weight graph and the merged feature graph are separated into three context features through the *Hadamard* product operation, and are then summed with $P_5$ to obtain the feature graph $F_6$ with rich context information. AF-FPN can effectively reduce the information loss caused by the reduction of feature channels through the AAM, allowing feature maps with rich context information to be obtained.

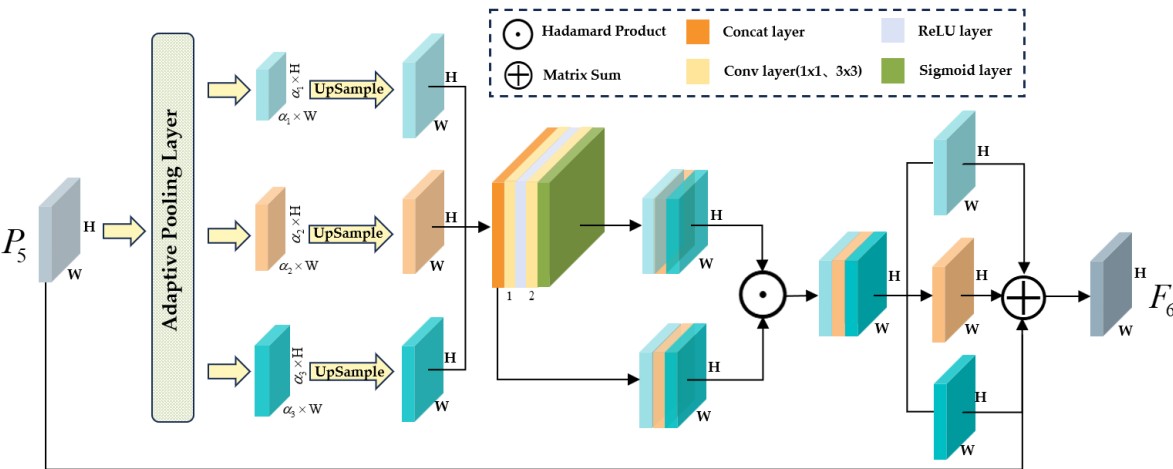

**Figure 3.** The specific structure of the AAM.

The structure of the FEM is shown in Figure 4. Its working process can also be divided into two parts. The first part is multi-branch convolution. This part includes a dilated convolution layer, a *BN* layer, and a *ReLU* function activation layer. The main function of the multi-branch convolution layer is the dilated convolution operation, which can adaptively learn the receptive fields of different sizes in each feature map according to the objects of different scales in the input image, thus improving the detection accuracy of the model for multi-scale objects. The dilated convolution kernels of the three branches are the same size ($3 \times 3$), while the expansion rate $\xi$ varies $(3, 5, 7)$. The expansion rate calculation formula is shown in Equations (1) and (2):

$$\xi_1 = t \times (k - 1) + 1 \tag{1}$$

$$\xi_n = t \times (k - 1) + \xi_{n-1} \tag{2}$$

where $\xi_n$ is the expansion rate of a branch ($n = 1, 2, 3$), $t$ is the convolution step size (where the step size is 1), and $k$ is the convolution kernel size.

During the processing, the internal elements of the dilated convolution kernel are distributed according to the expansion rate interval, which is different from the adjacent distribution of the internal elements of the standard convolution kernel, and the spatial size depends on the expansion rate. The expansion characteristics of the receptive field of the dilated convolution kernel do not cause a loss of resolution and coverage of the image. The relationship between the expansion rate of the dilated convolution kernel and the size of the receptive field is shown in Equation (3):

$$R_q = (\xi_n - 1) \times (k - 1) + k \tag{3}$$

where $R_q$ represents the size of the receptive field of a certain branch ($q = 1, 2, 3$).

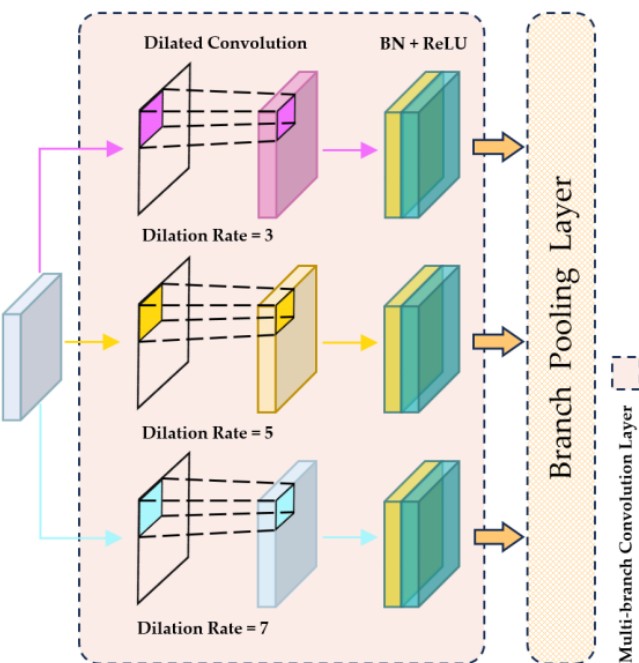

**Figure 4.** The specific structure of the FEM.

The second part is the branch pool layer. The object information in each parallel branch receptive field is fused by average pooling to avoid the need for additional parameters in the model calculation. This operation is used during the training of the model. By balancing the feature representation of different branches, the accuracy of multi-scale objects can be enhanced by a single branch during the test. The branch pool layer output and branch number expression are shown in Equation (4):

$$F_o = \frac{1}{B}\sum_{i=1}^{B} F_i \tag{4}$$

where $F_o$ is the output of the branch pooling layer fusing the characteristic information of each branch, $B$ is the number of branches to be fused in this layer ($B = 3$), and $F_n$ is the characteristic information of a branch ($i = 1, 2, 3$).

### 3.2.2. Prediction Head Based on Transformer Encoder

In order to improve the detection ability of the model to deal with tiny and high-density occluded objects, we added an additional prediction head on the basis of the three prediction heads of YOLOv7. This four-head structure, shown in Figure 3, can effectively alleviate the negative impact caused by drastic changes in object scale.

This prediction head combines a transformer encoder block [56] (T-encoder), where the structure of the T-encoder is shown in Figure 5. We introduce it into YOLOv7 to improve the capture ability of the model for different local information and explore the feature representation ability of the self-attention mechanism. T-encoders are stacked by $N$ encoders, where each encoder contains two sub-layers: a multi-head self-attention (MSA) layer and a multi-layer perceptron (MLP) layer. Each sub-layer is residually connected and arranged in sequence. The T-encoder mainly uses the self-attention mechanism of the MSA layer to capture the global dependencies between input patch sequences, obtain rich context semantic information, and enhance the model expression ability after further processing by MLP.

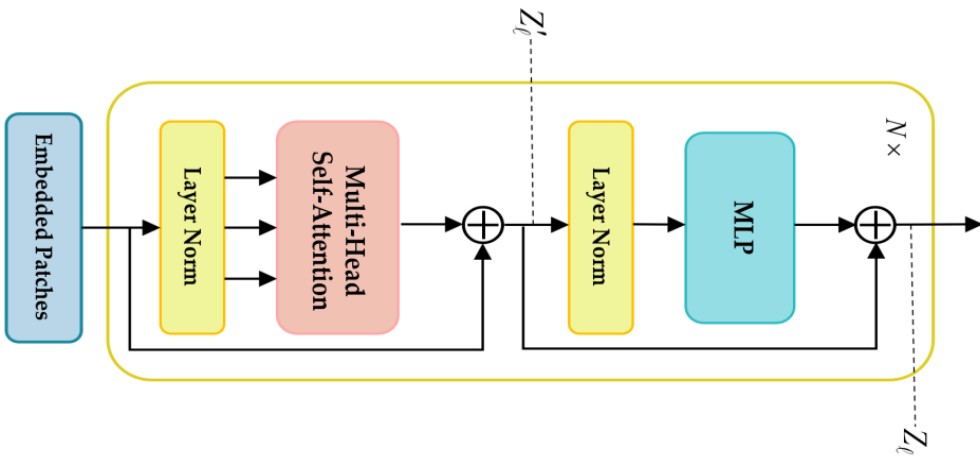

**Figure 5.** Structure of the transformer encoder.

Specifically, in each layer $\updownarrow$ of the encoder, the input sequence $Z_{\updownarrow-1}$ of the previous layer normalizes the input of each example on the feature dimension through layer normalization (LN), in order to prevent drastic changes in the value of each layer, thus improving the stability and generalization of the model in terms of feature representation. Then, the output of the previous layer LN passes through the MSA layer to obtain a new sequence $Z_{\updownarrow}$. Finally, the $Z_{\updownarrow}$ output, as a new input, is normalized and submitted to the MLP layer again to generate a set of the latest patch-embedded $Z_{\updownarrow}'$. Residual links are added between each sub-layer to enable the model to learn the residual function, addressing the problem of gradient disappearance in the processing flow of MSA. MLP is shown in Equations (5) and (6):

$$Z_{\ell} = \text{MSA}(\text{LN}(Z_{\ell-1})) + Z_{\ell-1}, \ \ell = 1 \cdots N \tag{5}$$

$$Z_{\ell}' = \text{MLP}(\text{LN}(Z_{\ell})) + Z_{\ell}, \ \ell = 1 \cdots N \tag{6}$$

The structure of MSA is shown in Figure 6. In the vision transformer [57] application, MSA measures the importance of the patch embedding in each part of the image through the self-attention mechanism, in order to capture the context semantic information between each part of the image. As MSA has multiple self-attention heads, it is called "multi-head self-attention", and it uses them to learn different parts of the input image. Each self-attention header in MSA contains three functions: query ($Q$), key ($K$), and value ($V$). These three functions belong to three complete connection layers. The self-attention mechanism calculates the attention scores of different patches embedded by these three functions, and utilizes these scores to determine the importance of different embedded patches. These scores are used for the weighted value of the trade-off, and the output represents the weighted sum of the self-attention mechanism. The MSA layer also includes a dot-product attention layer, a multi-head attention layer, concatenation, and a linear projection layer.

Specifically, the MSA layer first applies linear projection to the input sequence $z = x_1, x_2, x_3, \cdots, x_n$ embedded in the patch to generate corresponding matrices for $Q$, $K$, and $V$, where each input sequence has $D$ dimensions, the matrix is of size $D \times H$, and $H$ represents the number of heads. By learning the weight matrices $W_q$, $W_k$, and $W_v$, the values of $Q$, $K$, and $V$ can be obtained, as shown in Equation (7):

$$Q = zW_q, K = zW_k, V = zW_v \tag{7}$$

Then, the attention score is obtained by using the self-attention mechanism, $Q$, $K$ and $V$, as shown in Equation (8), where $Q$ and $K$ are dot products calculated and scaled by

the square root of their dimensions. The *softmax* function is applied to the last dimension corresponding to the attention score.

$$Attention(Q, K, V) = softmax\left(\frac{QK^T}{\sqrt{\frac{D}{H}}}\right)V \tag{8}$$

Finally, the output of the self-attention mechanism is cascaded through the head and transformed to the original dimension using linear projection. The final output of the MSA layer is realized through the learnable weight matrix $W_o$, and the final output is shown in Equation (9):

$$\text{MSA}(z) = concat(Attention_1, Attention_2, \cdots, Attention_H)W_o \tag{9}$$

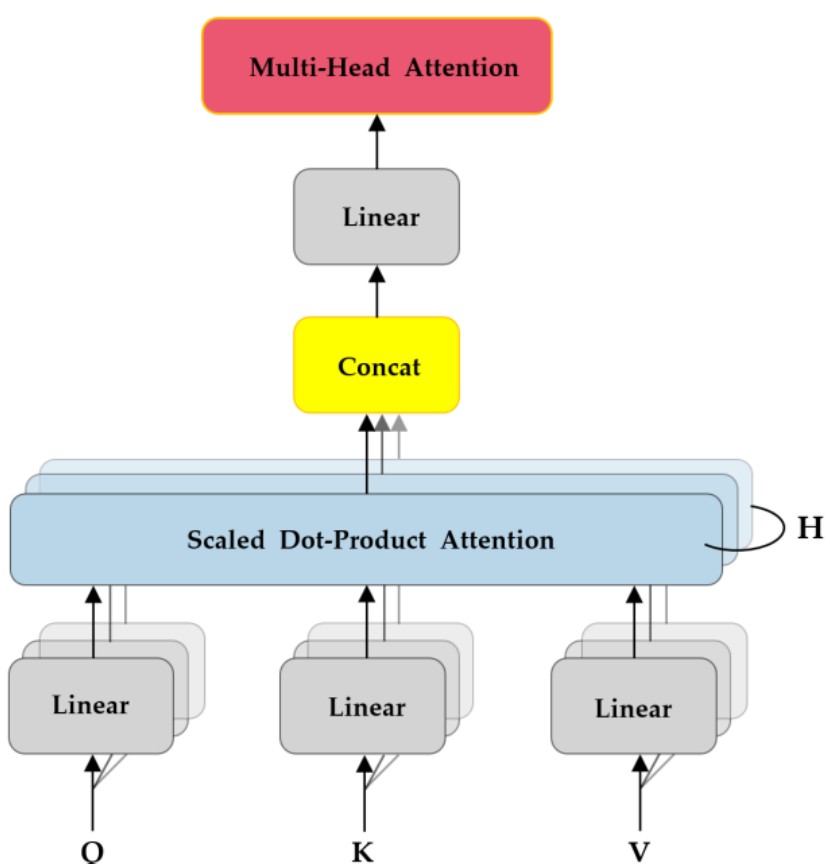

**Figure 6.** The structure of MSA.

The structure of MLP is shown in Figure 7. It consists of two linear transformation layers and a non-linear activation function layer. Each layer is fully connected, and the two linear transformations are separated by the non-linear activation function. The MLP layer is responsible for transforming the representation input from the MSA layer to a higher dimensional feature space. Specifically, first, in the first linear transformation layer, the learnable weight matrix $W_1$ is multiplied by the input embedded patch $X$, and the multiplied result is added to the offset $b_1$ to obtain $\text{MLP}_1$. The above process is shown in Equation (10):

$$\text{MLP}_1 = zW_1 + b_1 \tag{10}$$

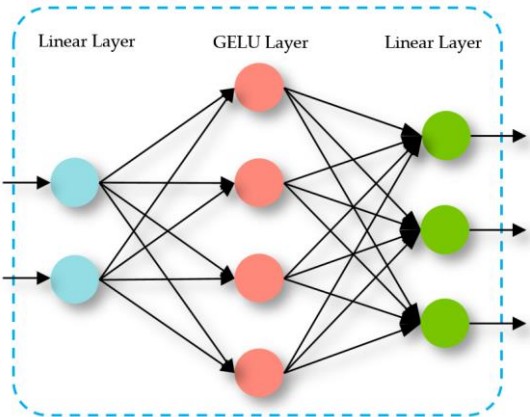

**Figure 7.** The structure of MLP. All layers are fully connected.

Second, after the output of the previous layer passes through the Gaussian error linear unit layer—that is, the *GELU* non-linear activation function layer—then $\text{MLP}_2$ is obtained. The above process is shown in Equations (11) and (12):

$$\text{MLP}_2 = GELU(\text{MLP}_1) \tag{11}$$

$$GELU(x) = xP(X \le x) = x\Phi(x) = x \cdot \frac{1}{2}\left[1 + erf\left(\frac{x}{\sqrt{2}}\right)\right] \tag{12}$$

where $P(X \le x)$ is the cumulative distribution function, representing the mean value of the Gaussian distribution at $x$, and $erf$ is the error function.

Finally, the output processed by the non-linear activation function layer enters the second linear transformation layer. The operation at this time is similar to that in the first linear transformation layer: $\text{MLP}_2$ is multiplied by the new learning weight matrix $W_2$, then the result is added to the new offset $b_2$ to obtain $\text{MLP}_3$. The above process is shown in Equation (13):

$$\text{MLP}_3 = \text{MLP}_2 W_2 + b_2 \tag{13}$$

As the T-encoder is composed of multiple encoders in series, the MLP layer is followed by the new MSA layer and MLP layer, and the series process is repeated $N$ times until the model learns the rich context information and complex abstract deep features of the input image. After the encoder block finishes the above operations, in order to generate the image representation $y$, the encoder selects the first embedded patch of sequence $z_N^0$ and performs layer normalization. This process is shown in Equation (14):

$$y = \text{LN}\left(z_N^0\right) \tag{14}$$

### 3.2.3. Improved Loss Function: SIoU

The location loss function CIoU of YOLOv7 considers the aspect ratio, distance, and overlapping area between the prediction box and the ground truth box, but fails to consider the vector angle between them, which greatly affects the convergence speed and detection performance during model training. Therefore, in order to improve the positioning accuracy and convergence speed when the model detects multi-scale and dense objects, we propose the use of a new loss function, SIoU [58], to optimize the model training loss.

Specifically, in addition to the characteristics of CIoU, SIoU can also consider the vector angle and direction between the prediction label and the ground truth label, and re-defines the penalty measurement when analyzing the regression model. Its comprehensive consideration of the training loss helps the model to be more robust and converge faster. Besides the distance cost $\Delta$, shape cost $\Omega$, and IoU cost, SIoU also introduces the angle cost

$\Lambda$. The angle cost $\Lambda$ is considered when calculating the distance cost $\Delta$. The definition of SIoU is shown in Equation (15):

$$L_{box\,SIoU} = 1 - IoU + \frac{\Delta + \Omega}{2} \tag{15}$$

The above four cost parameters determine the main functions of SIoU and are redefined by SIoU. The parameter relationship of SIoU is shown in Figure 8. The definitions of the various parameters are as follows: First, for the angle cost $\Lambda$, its definitions are shown in Equations (16)–(19):

$$\Lambda = 1 - 2 \times sin^2\left(arcsin\left(\frac{C_H}{\sigma}\right) - \frac{\pi}{4}\right) = cos\left(2 \times \left(arcsin\left(\frac{C_H}{\sigma}\right) - \frac{\pi}{4}\right)\right) \tag{16}$$

$$\sigma = \sqrt{\left(b_{c_x}^{gt} - b_{c_x}\right)^2 + \left(b_{c_y}^{gt} - b_{c_y}\right)^2} \tag{17}$$

$$C_H = max\left(b_{c_y}^{gt}, b_{c_y}\right) - min\left(b_{c_y}^{gt}, b_{c_y}\right) \tag{18}$$

$$\frac{C_H}{\sigma} = sin(\alpha) \tag{19}$$

where $C_H$ is the vertical height difference between the center of the ground truth box and the center of the prediction box, $\sigma$ is the distance between them, $arcsin\left(\frac{C_H}{\sigma}\right)$ is the angle $\alpha$, $\left(b_{c_x}^{gt}, b_{c_y}^{gt}\right)$ represents the center point of the ground truth box, and $\left(b_{c_x}, b_{c_y}\right)$ represents the center point of the predicted box. After analysis, it can be concluded that, when $\alpha$ is $\frac{\pi}{2}$ or 0, the angle cost $\Lambda$ is also 0. If $\alpha < \frac{\pi}{4}$, $\alpha$ should be minimized first; otherwise, $\beta$ should be minimized first.

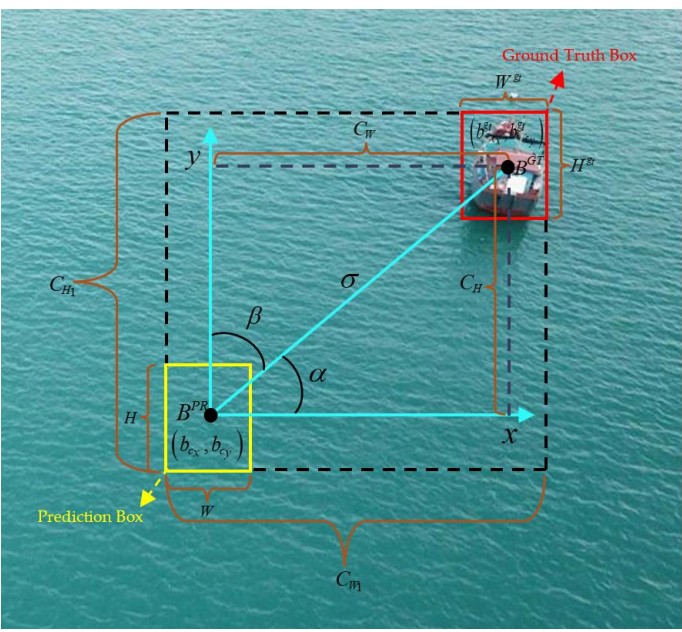

**Figure 8.** Parameter relationship of SIoU.

Second, the distance cost $\Delta$ is determined by the distance between the prediction box center $B^{PR}$ and the ground truth box center $B^{GT}$, and is affected by the angle cost $\Lambda$.

Considering the above definition of the new angle cost $\Lambda$, it is necessary to re-define the distance cost $\Delta$. Its definition is shown in Equations (20) and (21):

$$\Delta = \sum_{t=x,y} \left(1 - e^{-\gamma \rho_t}\right) = 2 - e^{-\gamma \rho_x} - e^{-\gamma \rho_y} \tag{20}$$

$$\rho_x = \left(\frac{b_{c_x}^{g^t} - b_{c_x}}{C_{W1}}\right)^2, \rho_y = \left(\frac{b_{c_y}^{g^t} - b_{c_y}}{C_{H1}}\right)^2, \gamma = 2 - \Lambda \tag{21}$$

It can be seen that the contribution of the distance cost decreases sharply with a decrease of $\alpha$ in the angle cost. However, when $\alpha$ approaches $\pi/4$, the contribution of the distance cost also increases. The value $\gamma$ indicates that, with an increase in angle, the time factor becomes the priority factor for calculating the distance value. $C_{W1}$ and $C_{H1}$ represent the width and height of the minimum boundary rectangle circumscribed by the prediction box and the ground truth box, respectively.

Furthermore, the shape cost $\Omega$ is defined as shown in Equations (22) and (23):

$$\Omega = \sum_{t=W,H} \left(1 - e^{-W_t}\right)^\theta = \left(1 - e^{-W_W}\right)^\theta + \left(1 - e^{-W_H}\right)^\theta \tag{22}$$

$$W_W = \frac{|W - W^{g^t}|}{max(W, W^{g^t})}, W_H = \frac{|H - H^{g^t}|}{max(H, H^{g^t})} \tag{23}$$

where $W$ and $H$ refer to the width and height of the predicted box, respectively, and $W^{g^t}$ and $H^{g^t}$ refer to the width and height of the ground truth box, respectively. As an important weight in the shape cost, the value of $\theta$ is used to determine the uniqueness of the shape cost in the dataset.

Finally, the IoU cost is defined as shown in Equation (24):

$$IoU = \frac{|area(B^{PR} \cap B^{GT})|}{|area(B^{PR} \cup B^{GT})|} \times 100\% \tag{24}$$

where $area(B^{PR} \cap B^{GT})$ is the intersection of the prediction box and the ground truth box, and $area(B^{PR} \cup B^{GT})$ is the union of the two. The ratio of the two constitutes the IoU cost. The specific relationship is depicted in Figure 9.

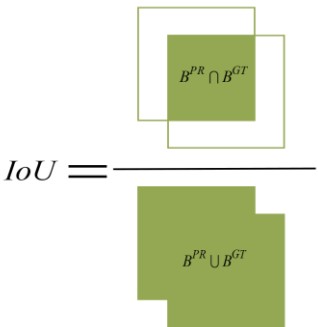

**Figure 9.** Relationship between IoU and prediction box and ground truth box.

## 4. Experiments and Results

### 4.1. Experimental Platform and Parameter Settings

Considering the limited computing resources of UAV platforms, and in order to maximize the data processing ability of the proposed mining model, for hardware we chose an Intel Core i9-12900kf CPU@3.20 GHz as the CPU. For software, we chose the Windows 11 operating system, PyTorch 2.0.0 as the deep learning framework, and CUDA 11.8 as

the GPU accelerator. The hardware and software settings of the experimental platform are detailed in Table 1.

**Table 1.** Experimental platform configuration.

| Parameter Name | Configuration |
| --- | --- |
| Operating System | Windows 11 |
| Integrated development environment | PyCharm |
| CPU | Intel Core i9-12900KF CPU@3.20 GHz |
| GPU | NVIDIA GeForce RTX 4090 (24G) |
| GPU accelerator | CUDA 11.8 |
| Deep learning frame | PyTorch 2.0.0 |
| Scripting language | Python 3.9.16 |
| Neural network accelerator | cuDNN v8.2.2 |

During model training, the AdamW optimizer was used to train ATS-YOLOv7. Considering the memory limitations of the experimental platform, the initialization parameters were set as follows. The cosine annealing strategy was used to update the learning rate, the weight decay regularization parameter was 0.0005, and the momentum was 0.937. The specific parameter settings used during training are given in Table 2.

**Table 2.** Training parameter configuration.

| Parameter Name | Configuration |
| --- | --- |
| Initial Learning rate | 0.001 |
| Cosine annealing hyperparameter | 0.2 |
| Neural network optimizer | AdamW |
| Size of input images | $640 \times 640$ |
| Batch size | 16 |
| Momentum | 0.937 |
| Weight decay | 0.0005 |
| Training epochs | 1000 |

*4.2. Data Set*

As the performance of a data-driven visual object detection model largely depends on the quality of data, the selection of an appropriate data set is particularly important. Models with different detection functions correspond to different data sets, and the requirements of UAV aerial image data corresponding to this paper are as follows:

➢ The scale of aerial data is large. UAVs collect a large amount of image data during the execution of aerial photography missions. If the detection model lacks a large amount of data support during the training period and there is not enough data to optimize a large number of parameters, the trained model will not have high generalization ability or robustness.

➢ The types of objects are comprehensive, and there may be many types of objects in aerial images. On one hand, in terms of the object scale, large, medium, and small objects will exist in the image; on the other hand, in terms of object types, aerial objects are varied, including objects such as ships, courts, cars, bridges, and so on. The consideration of comprehensive object types is conducive to training an efficient detection model.

➢ High-quality aerial images. The need for high-quality images can be viewed from two aspects. First, high-quality images provide good texture details and high resolution. Second, images collected in complex environments or with insufficiently performing cameras may suffer from complex backgrounds and blurred images, due to rain and snow weather, fuselage jitter, and so on. This type of image makes the model training closer to the real scene, so they can also be called high-quality. In this article, we also focus on the statistics of these types of images.



We selected several standard aerial image data sets, including DIOR, DOTA, UCAS-AOD, U-AIR [59], UAV-123 [60], UAVDT [61], NWPU VHR-10 [62], and OIRDS [63], and collected statistics on the data sets in the following five aspects: images, categories, instances, quality, and intra-class differences and inter-class similarities (Icdaics). The results are shown in Figure 10.

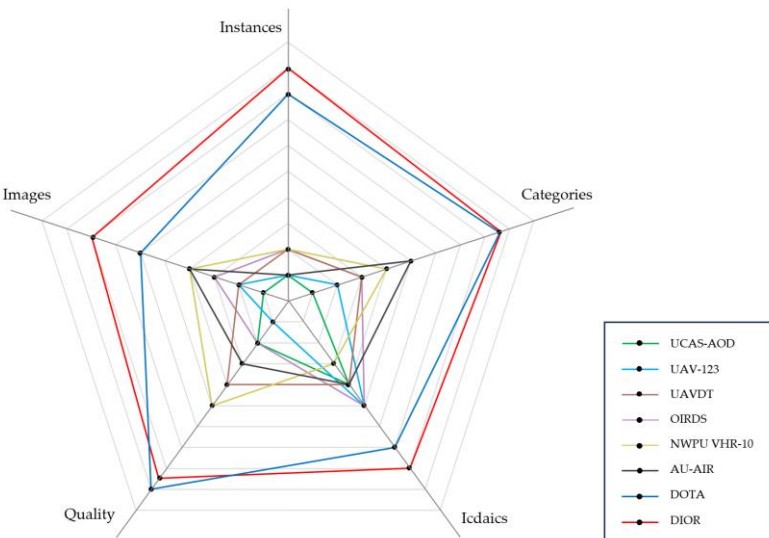

**Figure 10.** Multi-attribute comparison of eight aerial photo datasets.

From Figure 10, it can be seen that the DIOR dataset presented the best comprehensive performance in these five aspects. In particular, DIOR contains a large number of images (23,463), rich object types (20), and a large number of sample instances (192,472). Therefore, we chose it as the data basis for the model experiment in this paper. The dataset is divided into training set, verification set, and test set according to the ratio of 6:2:2. The statistics of DIOR, in terms of specific sample number and type, are shown in Figure 11.

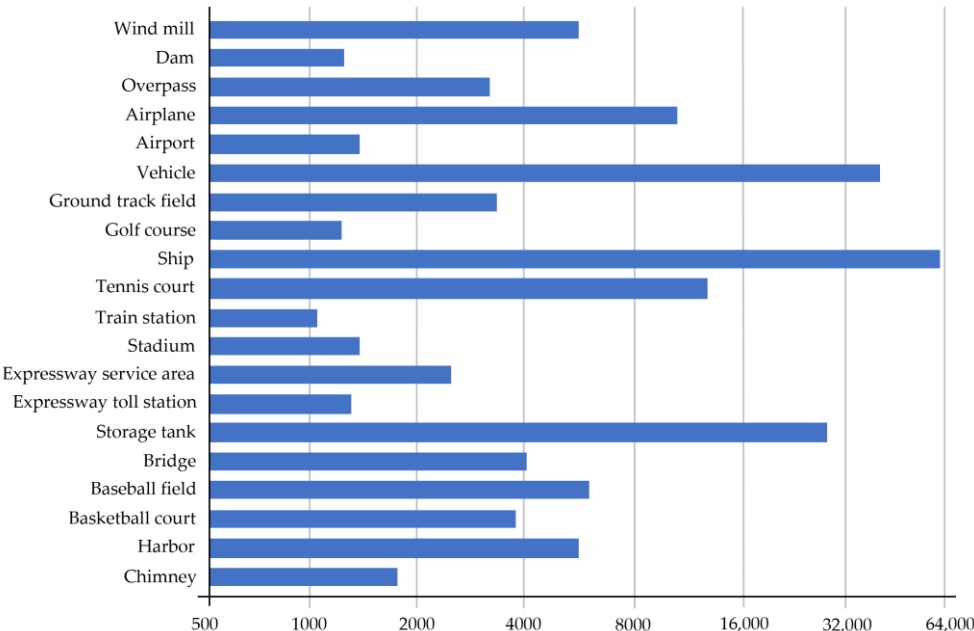

**Figure 11.** Data statistics (sample quantity and object type) for the DIOR dataset.

## 4.3. Evaluation Metrics

We selected a variety of indicators to evaluate the performance of the model and improve the modules, as follows: precision, recall, *F*1 score, *AP* (average precision), *mAP* (mean AP), and FPS (frames per second).

For the binary classification problem, the classification results regarding the combination of the ground truth box and the predicted box can be divided into four sample categories: true positive (TP), false positive (FP), true negative (TN), and false negative (FN). The confusion matrix of the classification results is presented in Table 3.

**Table 3.** Confusion matrix for the classification results.

| Labeled | Predicted | Confusion Matrix |
|---|---|---|
| Positive | Positive | TP |
| Positive | Negative | FN |
| Negative | Positive | FP |
| Negative | Negative | TN |

Precision (*P*) and recall (*R*) can be obtained from the data combination of the binary classification confusion matrix, defined as follows:

$$P = \frac{TP}{TP + FP} \tag{25}$$

$$R = \frac{TP}{TP + FN} \tag{26}$$

Precision represents the proportion of all true positive samples in the tested positive samples, while the recall rate represents the ability of the model to correctly detect positive samples. The *P*–*R* curve can be obtained by taking the precision and recall rate as the vertical and horizontal axes, respectively. In order to evaluate the comprehensive classification and recall ability of the classifier, the *F*1 score is obtained as the harmonic average of the above two indicators. The higher the *F*1 score, the more effective the test method is. The *F*1 score is defined as follows:

$$F1 = \frac{2 \times P \times R}{P + R} \tag{27}$$

*AP* refers to the average detection accuracy of various objects, and its size is the area bounded by the *P*–*R* curve and the horizontal axis. *mAP* measures the overall detection performance of the model for all kinds of objects, according to different thresholds (this paper takes the *mAP* of 0.5, with the *mAP* ranging from 0.5 to 0.95). It is the average of all object categories of *AP*, and is considered the best indicator for judging the comprehensive detection performance of a model. The specific definitions of *AP* and *mAP* are shown in Equations (28) and (29), respectively:

$$AP = \int_0^1 P(R) \times dR \times 100\% \tag{28}$$

$$mAP = \frac{1}{n} \sum_{i=1}^{n-1} AP(i) \times 100\% \tag{29}$$

where $P(R)$ represents the accuracy on the *P*–*R* curve, $n$ represents the object type of the set model detection, and $i$ represents a certain object to be detected.

FPS is an important index of the speed. It represents the number of frames per second that can be processed in the process of model reasoning. In order to verify the real-time performance of the model, the *FPS* of a model should ideally be greater than 30. *FPS* is composed of three parts: the initial image processing time (*Pre*), the model reasoning running time (*Inf*), and the non-maximum suppression time (*NMS*). Its definition is shown in Equation (30):

$$FPS = 1000/(Pre + Inf + NMS) \tag{30}$$

*4.4. Experiment and Results Analysis*

4.4.1. Precision–Recall Rate Experiment

We conducted a precision–recall experiment to verify the performance of the proposed method in two aspects: first, the impact of data set size and number of object categories on precision–recall and, second, the impact of the new algorithm, formed by combining the improved modules with YOLOv7, on the precision–recall rate. In this experiment, first of all, we combined three types of improved modules—namely, AF-FPN, T-encoder, and SIoU—with YOLOv7 to obtain the comparison models YOLOv7 + AF-FPN, YOLOv7 + T-encoder, and YOLOv7 + SIoU. Second, we trained and tested the above three models, the overall improved model ATS-YOLOv7, and the baseline model YOLOv7 on six different types of aerial photography data sets, then assessed the experimental data to draw *P–R* curves, as shown in Figure 12.

From the *P–R* curves, it can be seen that the performance of the five types of detection models differed on the different data sets. Overall, the detection ability increased with increasing data set size and object types. Among them, the five detection models had the best precision–recall rate on the DIOR data set, and the worst performance on the UCAS-AOD data set. After analysis, it was found that the data size and number of object types in the DIOR data set were the largest of the six data sets involved in the experiment, with 23,463 images and 20 object types, while the UCAS-AOD data set had only 2420 images and 2 object types. Therefore, by observing the experimental results, it can be concluded that comprehensive aerial photography data are very necessary for the development of a detection model, as this can enable the model to fully learn object features, capture key information, and adjust the training parameters to improve the model generalization and detection capabilities.

From the perspective of the performance of the improved module combined with the baseline model, ATS-YOLOv7 performed best in terms of *P–R* ratio on the six data sets, followed by YOLOv7 + AF-FPN and YOLOv7 + T-encoder, while YOLOv7 + SIoU performed slightly worse and YOLOv7—the original model—performed the worst. It can be seen that YOLOv7 was overall improved, in terms of feature information extraction, object location, resistance to redundant interference, and drastic scale changes, after integrating the three types of models, and the comprehensive performance of the model was greatly improved. Although YOLOv7 + AF-FPN, YOLOv7 + T-encoder, and YOLOv7 + SIoU improved over the baseline, to different degrees, they cannot guarantee comprehensiveness of the detection process and high accuracy of the detection results for complex aerial images captured by UAVs. Therefore, in general, the ATS-YOLOv7 method proposed in this paper performed well in multi-category aerial image data sets, presenting good detection performance when processing multi-type aerial objects.

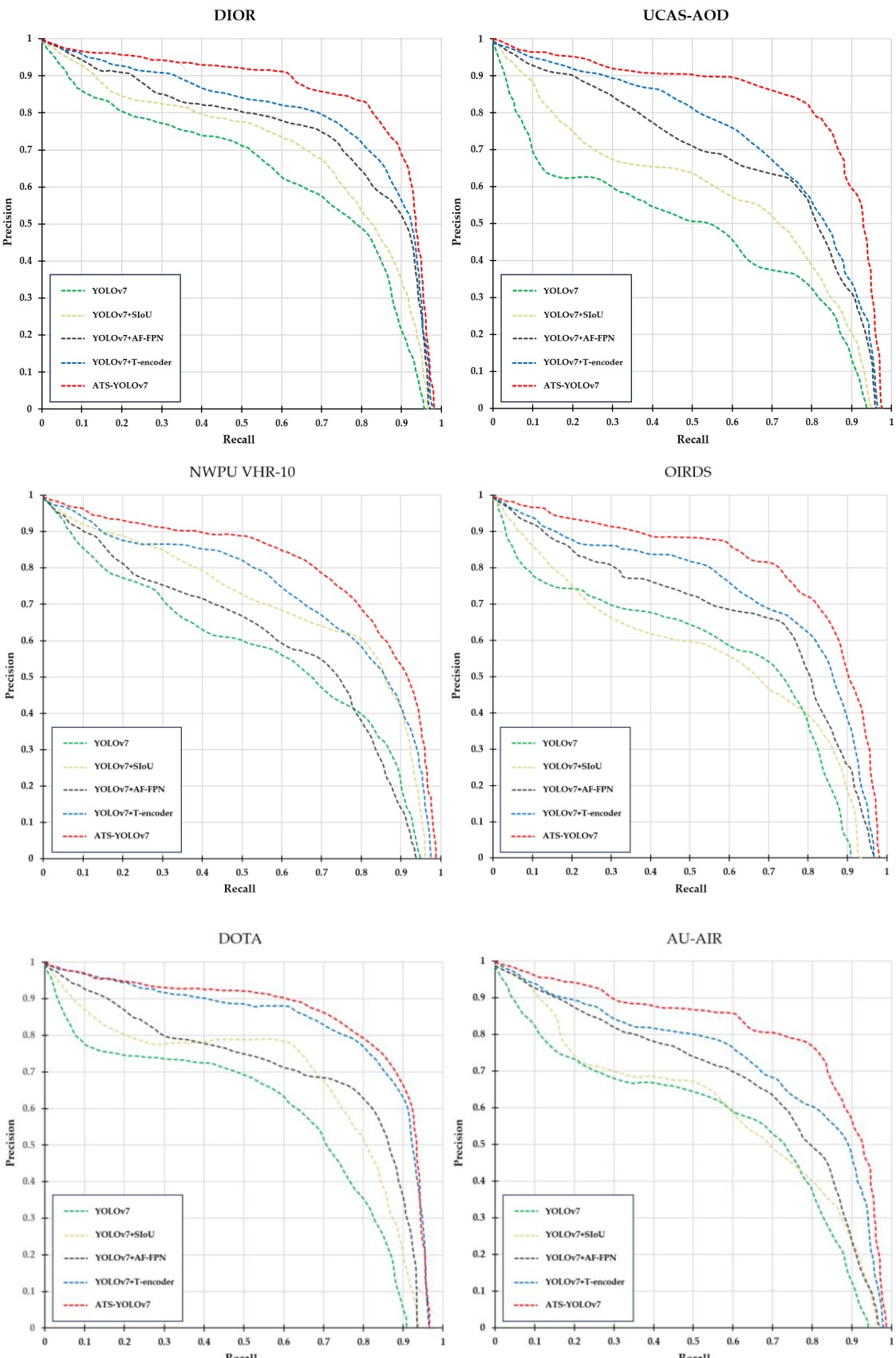

**Figure 12.** P–R statistical curves for the five comparison models YOLOv7 + AF-FPN, YOLOv7 + T-encoder, YOLOv7 + SIoU, YOLOv7, and ATS-YOLOv7 on six aerial photography data sets.

4.4.2. Ablation Experiment

In order to further verify the effectiveness and rationality of each improved module when combined with YOLOv7, we conducted ablation research on the DIOR dataset. The baseline model was YOLOv7 and each progressive module was trained based under the same experimental settings. The respective contributions of the models are intuitively shown in Table 4, in which "√" indicates that the improved module was added, while a blank space indicates that the module was not selected.

**Table 4.** Ablation results considering the improved modules. The data unit is percentage (%).

| Baseline | AF-FPN | T-Encoder | SIoU | Dataset | Parameters | FPS | $AP_t$ | $AP_s$ | $AP_m$ | $AP_l$ | $mAP@0.5$ | $mAP@0.5:0.95$ |
|---|---|---|---|---|---|---|---|---|---|---|---|---|
| YOLOv7 | | | | DIOR | 71.4 M | 110 | 59.1 | 63.3 | 79.1 | 83.7 | 73.1 | 58.7 |
| | √ | | | | 72.2 M | 100.7 | 70.3 | 73.1 | 82.2 | 85.5 | 74.3 | 63.3 |
| | | √ | | | 74.4 M | 109 | 72.1 | 75.1 | 80.7 | 85.1 | 75.8 | 65.8 |
| | | | √ | | 71.4 M | 100.3 | 62 | 72.7 | 78.1 | 79.3 | 71.3 | 60.1 |
| | √ | √ | | | 75.7 M | 90.4 | 66.3 | 78.2 | 88.3 | 93.9 | 80.5 | 66.8 |
| | √ | | √ | | 75.2 M | 92.6 | 60.1 | 74.3 | 85.9 | 90.6 | 79.7 | 64.7 |
| | | √ | √ | | 75.4 M | 96.1 | 63.3 | 77.5 | 85.1 | 88.1 | 73.1 | 62.1 |
| | √ | √ | √ | | 80.9 M | 94.2 | 72.8 | 79 | 90.3 | 94 | 81.3 | 70.3 |

During the experiment, we used a variety of evaluation methods according to the $AP$ and $mAP$, including $AP_t$, $AP_s$, $AP_m$, $AP_l$, $mAP@0.5$, and $mAP@0.5:0.95$. In particular, $AP_t$, $AP_s$, $AP_m$, and $AP_l$ represent the average precision for tiny, small, medium, and large targets, respectively, while $mAP@0.5$ and $mAP@0.5:0.95$ represent the average value of $AP$ for all scale objects when the SIoU threshold was set at 0.5 and the average value of $AP$ for all scale objects when the step size was 0.05 from 0.5 to 0.95, respectively. These two types of $mAP$ are increasingly difficult for each detection model, and they can be used as an important indicator to measure the comprehensive detection capability of a model. For convenience of expression in the following, we denote YOLOv7 + AF-FPN by ①, YOLOv7 + T-encoder by ②, YOLOv7 + SIoU by ③, and the other models by ④, ⑤, ⑥, and ⑦ in turn.

From Table 4, the following observations can be made. First, compared to YOLOv7, the models with single improvement modules presented improvements in all six indicators, with ② having the highest increases in $AP_t$, $AP_s$, $mAP@0.5$, and $mAP@0.5:0.95$ (13%, 11.8%, 9.7%, and 7.1% higher, respectively). This indicates that YOLOv7 is more sensitive to the features of tiny and small objects when facing drastic variations in object scale, and its detection ability is improved after adding the T-encoder prediction head. Although the parameters were increased by 3 M, this did not affect the real-time requirements of the network. Second, in the models with two improved modules, the six indicators of ④ were the most improved over those of YOLOv7, which were 7.2%, 14.9%, 9.2%, 10.2%, 7.4%, and 8.1% higher, respectively. Due to the new AF-FPN architecture of the neck of the model and the additional detection heads, the parameters increased by 4.3 M, but the network reduced the information loss in the convolution process and strengthened the feature expression, which was conducive to the detection of complex object types. Finally, it can be seen that the baseline model YOLOv7 had the largest difference when compared with the comprehensive improvement ⑦ (ATS-YOLOv7), being 13.7%, 15.7%, 11.2%, 10.3%, 8.2%, and 11.6% lower, respectively. From the above analysis, it can be seen that YOLOv7 could detect and recognize multi-scale objects and densely occluded objects well after adding the AF-FPN module, T-encoder module, and optimizing the SIoU loss in training. Although ⑦ presented the largest increase in parameter quantity, it also managed to meet the real-time requirements in accordance with the FPS standard. The various kinds of modules can play a coordinated role in improving YOLOv7, verifying that the comprehensively improved YOLOv7 model proposed in this article is scientific and efficient.

Based on the above analysis, in order to highlight the advantages of ATS-YOLOv7 over YOLOv7 and, more specifically, to analyze the differences between the two models in terms of accuracy for various objects, we used the confusion matrix to assess their detection results. The confusion matrix is shown in Figure 13. The horizontal axis in the figure represents the ground category labels, the vertical axis represents the predicted category labels, and the cross term is the result of their classification accuracy. In order to observe the visual detection results, we abbreviated the name of each object in the data set, as detailed in Table 5. In order to save the detection time occupied by multiple similar objects, we selected 10 representative objects from the 20 object categories of DIOR for testing.

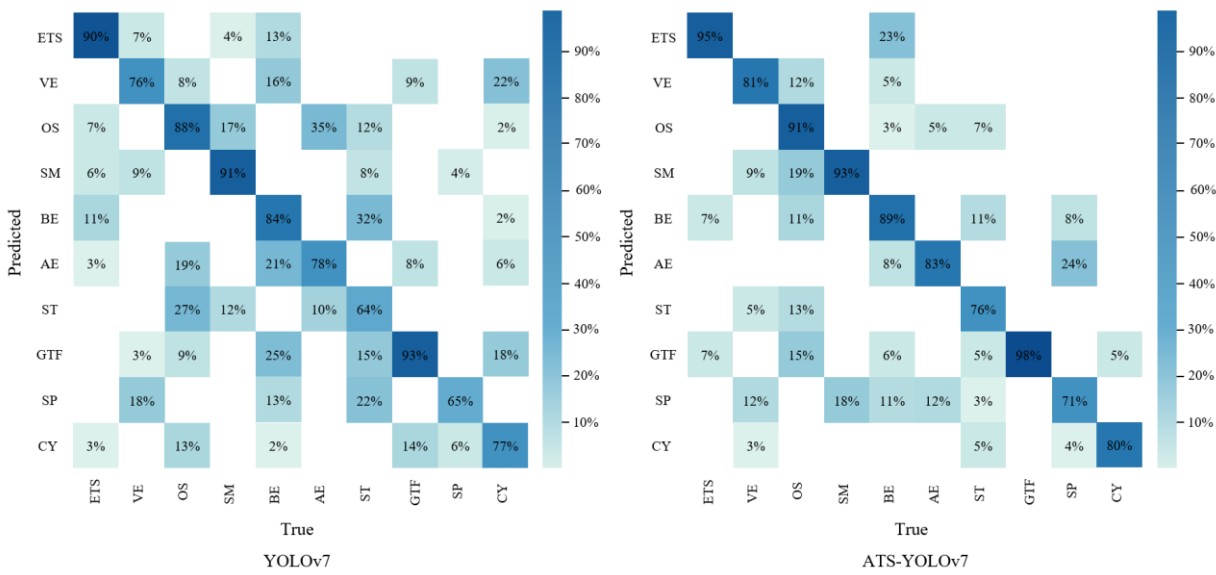

**Figure 13.** The confusion matrices for YOLOv7 and ATS-YOLOv7 on 10 types of challenging objects.

**Table 5.** Abbreviation of object categories.

| Object Name | Airport | Train station | Tennis court | Wind mill | Bridge | Expressway service area |
|---|---|---|---|---|---|---|
| Abbreviation | AT | TS | TC | WM | BE | ESA |
| Object Name | Harbor | Stadium | Basketball court | Golf course | Ship | Expressway toll station |
| Abbreviation | HR | SM | BC | GC | SP | ETS |
| Object Name | Vehicle | Airplane | Baseball field | Storage tank | Dam | Ground track field |
| Abbreviation | VE | AE | BF | ST | DM | GTF |
| Object Name | Chimney | Overpass | - | - | - | - |
| Abbreviation | CY | OS | - | - | - | - |

Overall, among the detected object categories, ATS-YOLOv7 had a higher classification accuracy for most objects when compared to YOLOv7, and the maximum difference in accuracy between the two was 12%. The overall false detection rate of ATS-YOLOv7 was lower than that of YOLOv7. Specifically, the performance difference between the two models in terms of large object detection (e.g., ETS, GTF, and SM) was not obvious, and both had high classification ability. In terms of medium objects (BE and OS), ATS-YOLOv7 had 5% and 3% higher accuracy, respectively. In terms of small objects (VE, CY, and AE), ATS-YOLOv7 had 5%, 3%, and 5% higher accuracy, respectively. The two presented the greatest difference for tiny objects (ST and SP), where ATS-YOLOv7 had 12% and 6% higher accuracy than YOLOv7, respectively. These results indicate that YOLOv7 with three detection heads is not sensitive to tiny objects, while the ATS-YOLOv7 model with AF-FPN and T-encoder has

improved attention due to the feature layer and can fully mine the feature information of objects, allowing it to perform better on more difficult objects.

### 4.4.3. Comparison with the State-of-the-Art

In this section, we compare the performance of the proposed method with a variety of SOTA visual detection models, including one- and two-stage models, on the DIOR data set, in order to illustrate the advantages of the ATS-YOLOv7 algorithm in terms of the current aerial image detection task. The chosen experimental models included RetinaNet [64], Scaled-YOLOv4 [65], YOLOv5 [22], TPH-YOLOv5 [66], HR-Cascade++ [67], O²DETR [68], DBAI-Net [69], YOLOv7 [23], and GLENet [70]. The specific test results for each model are given in Table 6. Each object is marked according to the scale size to display the differences in detection results, such as large (l), medium (m), small (s), and tiny (t).

**Table 6.** Statistical results of *AP*, *F*1 Score, and *mAP* for ATS-YOLOv7 and various SOTA models.

| Object Category | Method (*AP/F*1) | | | | | | | | | |
|---|---|---|---|---|---|---|---|---|---|---|
| | GLENet | Scaled-YOLOv4 | YOLOv5 | TPH-YOLOv5 | HR-Cascade++ | O²DETR | DBAI-Net | RetinaNet | YOLOv7 | Ours |
| AT (m) | 89/83 | 84/77 | 64/69 | 85/79 | 67/74 | 74/69 | 80/76 | 79/72 | 89/**86** | **90**/83 |
| TS (l) | 88/85 | 87/75 | 68/71 | **90/87** | 83/72 | 78/71 | 71/65 | 84/76 | 84/79 | 87/83 |
| TC (m) | **92**/83 | 71/64 | 61/70 | 84/80 | 62/70 | 86/84 | 77/72 | 75/70 | 90/84 | 89/**86** |
| WM (l) | 75/71 | 76/72 | 70/75 | 73/78 | 75/66 | 88/85 | 87/83 | 63/67 | **89/83** | 78/76 |
| BE (m, l) | 87/80 | 60/55 | 62/70 | 64/70 | 78/81 | 76/80 | 70/68 | 70/59 | 76/71 | **89/82** |
| ESA (l) | **93/88** | 79/75 | 65/78 | 82/80 | 64/69 | 73/60 | 89/84 | 89/75 | 87/83 | 90/85 |
| HR (l) | 87/71 | 80/73 | 67/58 | 64/56 | 74/78 | 79/75 | 74/78 | 78/71 | 74/78 | **88/82** |
| SM (l) | **91**/87 | 72/70 | 70/78 | 75/78 | 86/79 | 81/76 | 90/84 | 90/86 | 87/84 | 83/**88** |
| BC (m, l) | 92/67 | 76/71 | 61/53 | 66/57 | 78/81 | 88/81 | 82/78 | 67/71 | 85/79 | **95/87** |
| GC (l) | 89/83 | 85/78 | 61/70 | 84/72 | 63/68 | 74/62 | 85/81 | 76/84 | **90/86** | 87/81 |
| SP (t) | 59/54 | 70/64 | 51/60 | 58/4s9 | 47/61 | 66/58 | 64/58 | 48/61 | 67/62 | **71/65** |
| ETS (l) | 93/81 | 90/86 | 57/66 | 78/67 | 73/60 | 94/86 | 92/88 | 88/84 | 92/86 | **95/90** |
| VE (t, s) | 63/75 | 79/66 | 81/76 | 82/75 | 72/**78** | 80/77 | 76/72 | 58/64 | 74/69 | **84**/78 |
| AE (s) | 89/63 | 71/59 | 46/61 | 71/67 | 52/74 | 80/73 | 75/70 | 80/76 | **87/82** | 83/77 |
| BF (m) | 88/80 | 84/70 | 75/68 | 84/81 | 88/84 | **89/85** | 86/82 | 87/82 | 85/79 | 89/84 |
| ST (t, s) | 74/63 | 48/62 | 64/58 | 54/49 | 61/68 | 57/69 | 64/59 | 45/57 | 57/54 | **76/73** |
| DM (m) | 91/80 | 88/75 | 69/78 | 71/84 | 65/82 | 83/81 | 80/75 | 87/76 | 87/82 | **92/86** |
| GTF (l) | 96/91 | 84/88 | 66/80 | 94/85 | 90/92 | 86/92 | 88/85 | 74/87 | 89/86 | **97/94** |
| CY (t, s) | 71/68 | 75/68 | 54/68 | 61/74 | **80**/72 | 78/68 | 59/64 | 52/61 | 75/69 | **80/75** |
| OS (m) | 90/76 | 79/82 | 65/73 | 87/81 | 61/64 | 90/**85** | 82/78 | 89/74 | 83/77 | **91**/85 |
| *mAP* | 85 | 77 | 64 | 75 | 71 | 80 | 79 | 74 | 82 | **87** |

From the experimental results, it can be seen that the detection results of the various models for large- and medium-sized objects were generally higher than those for small and tiny objects, indicating that small and tiny objects are extremely challenging in the field of object detection. Our proposed model, ATS-YOLOv7, presented the best overall performance among various visual models (*mAP* of 87%, *F*1 score overall higher than other models), being especially superior to other models for small and tiny objects. Specifically, the 10 models had high detection results for large- and medium-sized objects, such as stadiums, bridges, sports fields, airports, and so on. Among them, GLENet obtained the best results (*mAP* of 90%), 26% higher than the worst YOLOv5 model. The difference between ATS-YOLOv7 and the best-performing model was only 0.4%, indicating that the method proposed in this article also has efficient performance for large- and medium-sized objects. However, for small and tiny objects, such as cars, ships, and chimneys, ATS-YOLOv7 presented the highest score compared to other models, being 4.9% higher in accuracy than the second scoring model YOLOv7. These results demonstrate that our proposed AF-FPN module, T-encoder module, and SIoU can efficiently enhance the detection of medium and large objects by improving upon YOLOv7, and that the proposed

model demonstrates superior performance when considering tiny and challenging objects in UAV aerial images.

Based on the above experimental results, we visualized the detection results of four typical challenging objects when using RetinaNet, Scaled-YOLOv4, YOLOv5, TPH-YOLOv5, HR-Cascade++, O$^2$DETR, DBAI-Net, YOLOv7, GLENet, and ATS-YOLOv7 models, as shown in Figure 14.

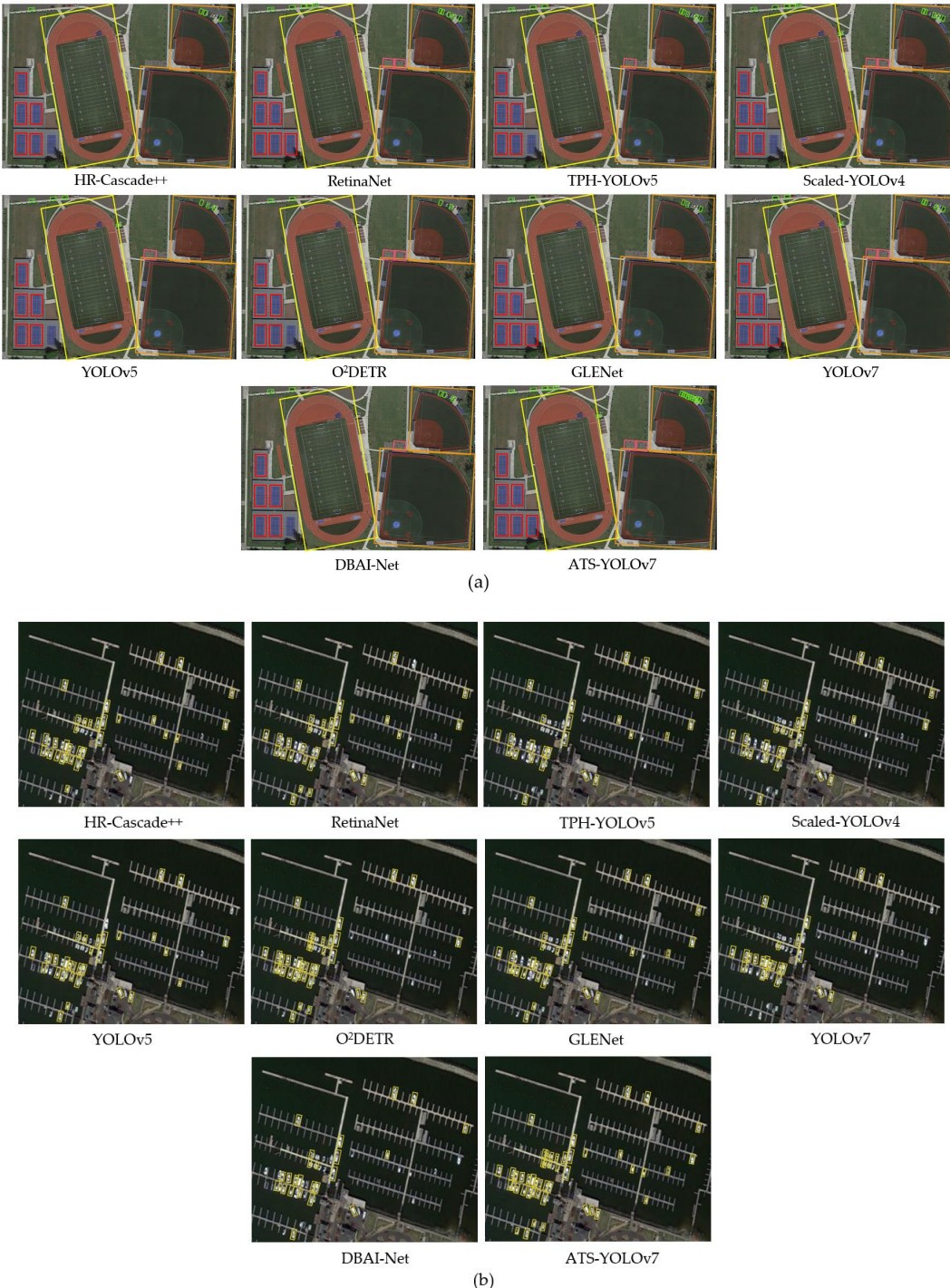

**Figure 14.** *Cont.*

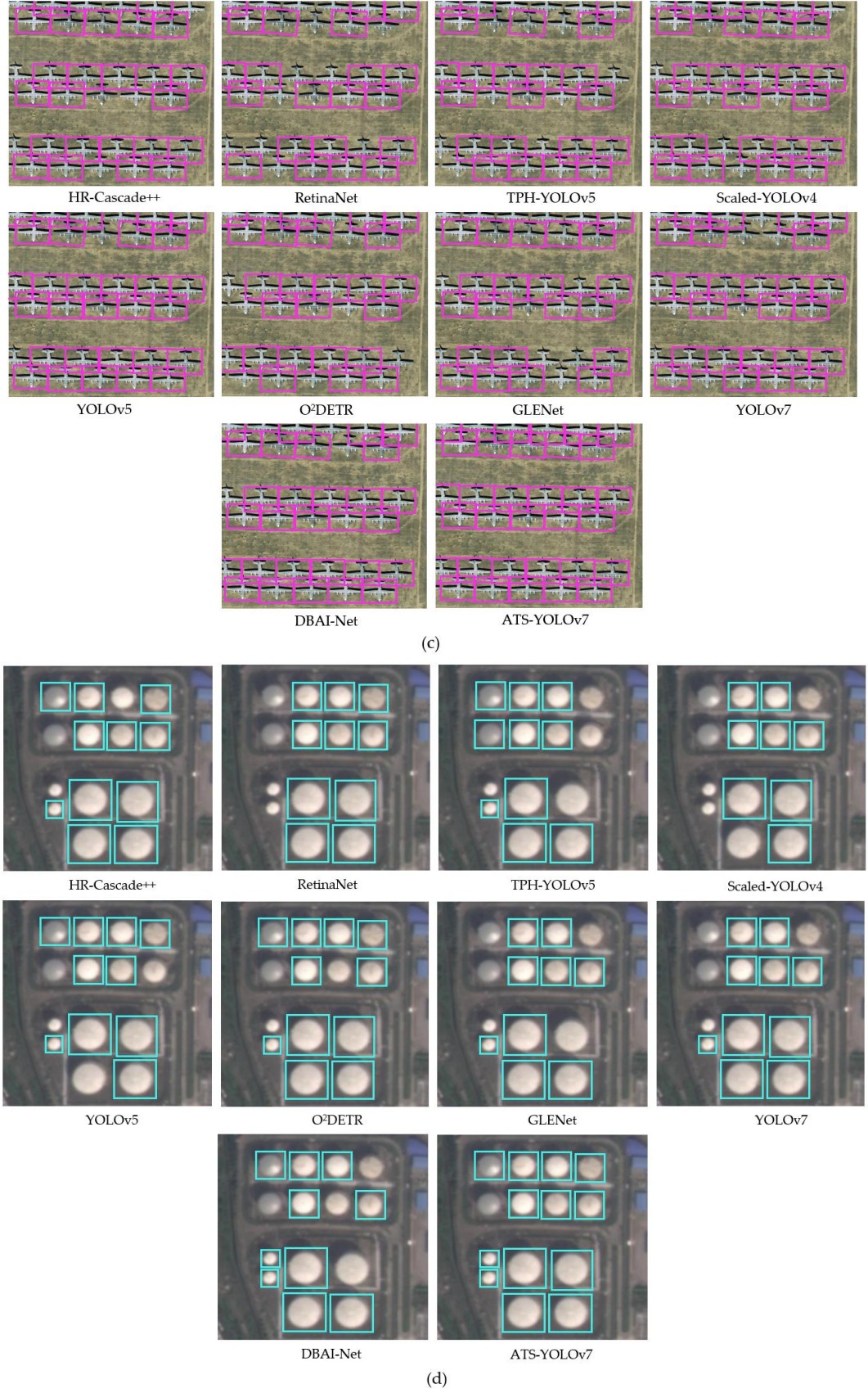

**Figure 14.** (**a**–**d**) The detection results of various models for multi-scale objects, tiny objects in complex scenes, densely arranged objects, and objects with fuzzy backgrounds.

Figure 14a shows the results of each model for multi-scale targets in a large scene. The objects in the picture are a ground track field, a basketball court, a tennis court, a golf course, and a car. Their dimensions vary from large to small, and their positions are scattered. It can be seen from the figure that ATS-YOLOv7 had the fewest misdetected objects, while HR-Cascade++ had the most misdetected objects, and the rest of the models had varying degrees of misdetection. This can be explained by the adaptive feature enhancement network, AF-FPN, of ATS-YOLOv7 helping the network reduce the negative impact on the network when multiple objects at varying scales change violently in large scenes, helping the detection performance of the network to remain stable.

Figure 14b shows the results of each model for tiny-scale ship targets. Observing the detection results, as ATS-YOLOv7 adds a detection route based on the T-encoder module to form a four-head detection structure, the network had a strengthened ability to capture the context information of the object area, making it more sensitive to the characteristics of tiny objects. Therefore, its detection results were the best, while the other models seriously missed some objects.

Figure 14c shows the results of the various models for densely arranged aircraft targets. From the perspective of prediction, the detection rate of YOLOv7 was the lowest, while the detection rate of ATS-YOLOv7 was the highest. Statistically, it was found that the detection result of ATS-YOLOv7 was about 10% better than that of YOLOv7. Therefore, after adding the AF-FPN, T-encoder, and SIoU functions to ATS-YOLOv7, the regression positioning and feature extraction abilities of the model were greatly improved. In the face of densely arranged objects with a similar background, it can maintain a high robustness and detection level.

Figure 14d shows the results of each model for storage tank targets with a fuzzy background. It can be seen that ATS-YOLOv7 detected and recognized storage tanks in fuzzy states better than the other models, as a result of the self-attention mechanism and adaptive feature strengthening mechanism in ATS-YOLOv7. However, in this state, if the storage tank is similar to the background, ATS-YOLOv7—like most other models—had trouble detecting the storage tank. In the future, we hope to further study TPH-YOLOv5 to improve our model architecture in this regard.

In general, for objects with a large scale, significant features, and a clear background, the networks could easily extract their rich feature information, then classify and locate them more accurately. However, for those complex objects with a tiny scale and fuzzy background, the network detection task becomes very difficult. Thus, the current aerial object detection network model not only relies on training on a large volume of data, but also requires comprehensive aerial data types and an advanced model architecture. On this basis, the model generalization and detection ability can be better improved.

The following shows some of the test results of our method on other aerial datasets. As shown in Figure 15, in general, ATS-YOLOv7 still has a high detection performance for challenging objects on other datasets. A few cases of missed detection are caused by similar backgrounds and small differences in the number of pixels, which is also what we need to pay attention to in the future.

Above, the detection accuracy of each model for objects in different scenes and types was analyzed in detail. Next, real-time experiments and result statistics were carried out, according to the FPS value obtained by each model. The real-time performance of a model is an important criterion for evaluating its comprehensive performance. The experimental results are shown in Figure 16.

As each network did not use the images in the DIOR validation set during training, we selected some images from the validation set for experiments, in order to make the experimental results more consistent with a real scenario. It can be seen, from the figure, that ATS-YOLOv7, YOLOv7, and TPH-YOLOv5 were the fastest in terms of reasoning speed, reaching 94 fps, 110 fps, and 105 fps, respectively. This result indicates that these three models had the highest speed advantage for the objects in the validation set. However, in terms of accuracy (see Table 6), ATS-YOLOv7 reached 87%, 5% higher than YOLOv7. As for the reasoning speed of ATS-YOLOv7, although presented no advantage over YOLOv7 and ATS-YOLOv7, it had little difference in comparison with them. As the object detection task of an UAV aerial image is an application project, we should consider the reasoning speed and detection accuracy comprehensively when exploring the real-time nature of the model, seeking a balance between the two. Based on this, from a comprehensive point of view, ATS-YOLOv7 has both accuracy and speed advantages and, so, is the most applicable model among these models.

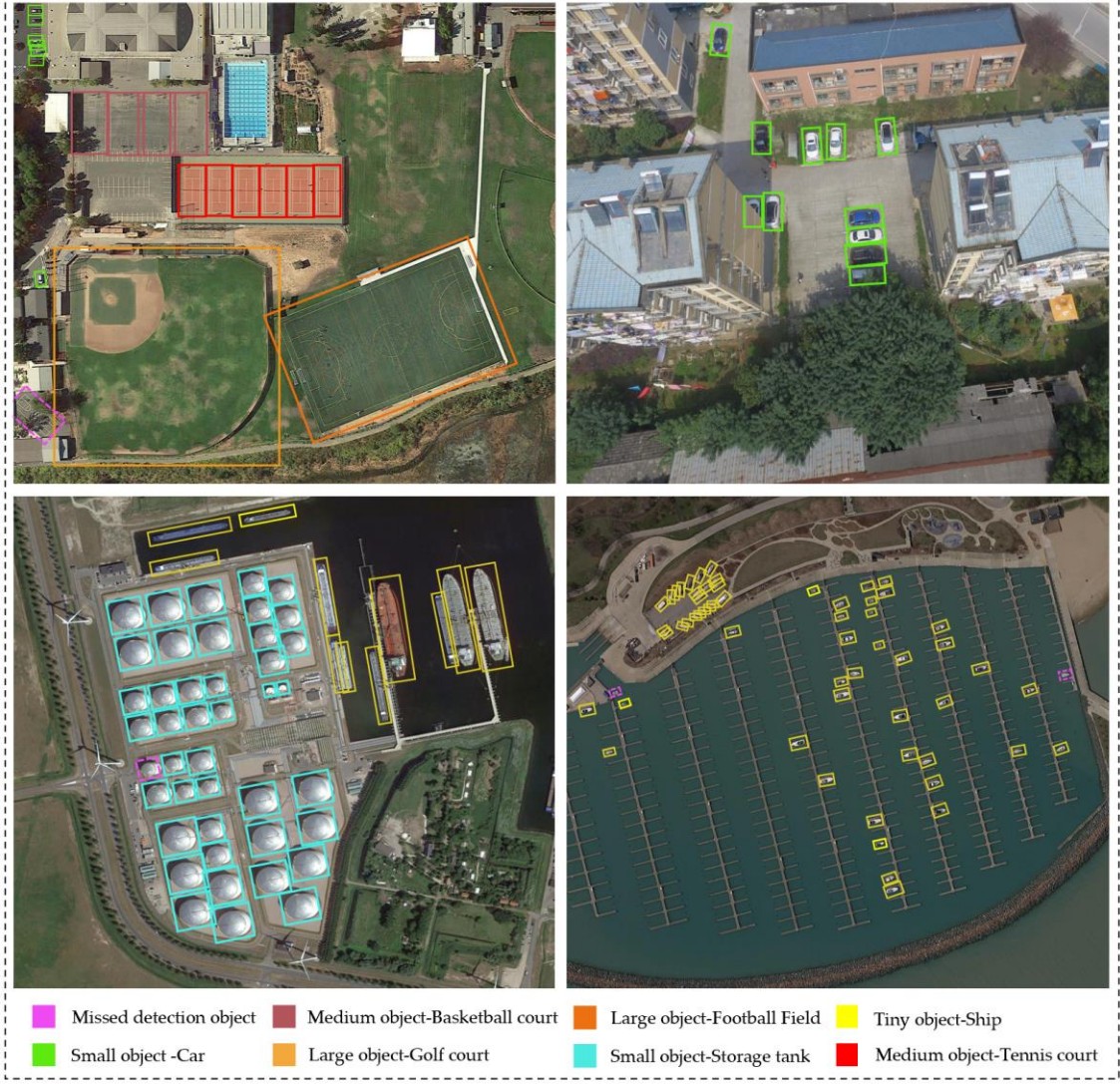

**Figure 15.** The four pictures respectively show the detection of multi-scale, occluded, densely arranged, and tiny objects by ATS-YOLOv7 on other datasets.

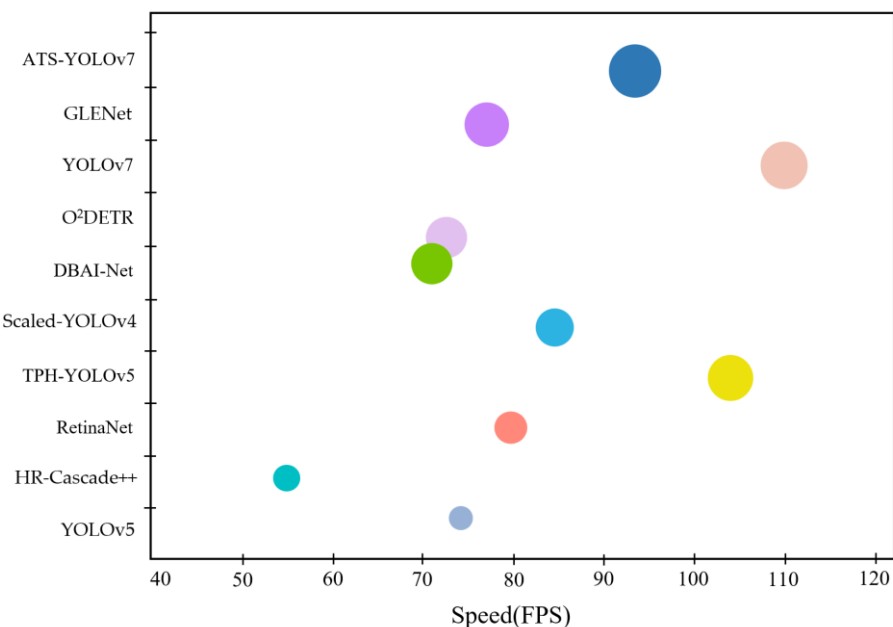

**Figure 16.** Real-time statistics of 10 detection models on the DIOR validation set.

## 5. Conclusions

This article proposed an improved YOLOv7 based solution strategy, called ATS-YOLOv7, to address the challenges faced by existing detection models in drone aerial image object detection tasks, with a focus on solving the problem of multi-scale objects. First, the adaptive feature enhancement pyramid network AF-FPN is utilized to reduce the loss of deep feature information and improve the model's feature sensitivity, thereby enhancing the model's detection speed and accuracy for multi-scale objects. Second, the additional detection head of the transformer encoder block enhances the model's ability to capture global information and enhances its ability to detect objects at a very small scale. Moreover, an SIoU loss with angle regression ability is proposed in this paper, which improves the regression ability and speed of model detection. Finally, through a series of experiments on aerial image datasets, we verified that ATS-YOLOv7 has the best detection accuracy (*mAP* of 87%) while maintaining a real-time image processing ability (94.2 FPS). In addition, we have also detected various typical targets on other datasets using ATS-YOLOv7 and achieved good results. Therefore, overall, the method proposed in this paper has high reliability and generalization ability when performing challenging target detection tasks in drone aerial images, and can effectively apply to real scenarios for drone aerial image target detection.

We discovered some areas in which improvements can be made through future research. First, the number of types of aerial objects needs to be increased, such that the trained network can better meet the object requirements of UAV aerial photography. Second, the scale of aerial images in bad weather needs to be compensated, in order to allow the model to better respond to complex and changeable application scenarios. Third, the network structure needs to be further adjusted and the number of parameters should be reduced, in order to realize a lightweight model. Fourth, in the future, we will study unsupervised models to fully utilize data and use unsupervised models to reduce complex data annotation work and improve our development efficiency.

**Author Contributions:** Conceptualization, H.Z., F.S. and X.H.; methodology and software, H.Z. and F.S.; validation and formal analysis, H.Z., D.Z. and Z.Z.; resources and data curation, X.H. and W.C.; writing—original draft preparation, review, and editing, H.Z., F.S. and X.H.; project administration and funding acquisition, F.S. and S.B. All authors have read and agreed to the published version of the manuscript.

**Funding:** This research was funded by the National Natural Science Foundation of China (grant number: 61671470).

**Data Availability Statement:** The data that support the findings of this study are available from the corresponding author Faming Shao (shaofaming@126.com) upon reasonable request.

**Conflicts of Interest:** The authors declare no conflict of interest.

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
