# Peer review of "ATS-YOLOv7: A Real-Time Multi-Scale Object Detection Method for UAV Aerial Images Based on Improved YOLOv7"

_electronics, doi:10.3390/electronics12234886_

Round 1

Reviewer 1 Report

Comments and Suggestions for Authors

Please find the below minor revision comments 

1.The literature review section appears to be somewhat limited. Expanding this section to include more recent and relevant studies could provide a better context for the research and demonstrate its significance in the current field.

2.The way data is presented in the manuscript could be improved. This includes clearer figures, tables, and possibly the addition of supplementary material to aid in the understanding of the research findings.

3.The results and discussion sections could be made more comprehensive. Providing a deeper analysis of the findings and discussing their implications in a broader context would add value to the manuscript.

4.Certain technical aspects, such as the accuracy of equations, the precision of technical terms, and the consistency of technical descriptions throughout the manuscript, could be revised for clarity and correctness.

Author Response

Response to Reviewer 1 Comments

Point 1: The literature review section appears to be somewhat limited. Expanding this section to include more recent and relevant studies could provide a better context for the research and demonstrate its significance in the current field.

Response 1: According to your suggestion, we have added and updated some latest relevant studies in the relevant part of the literature review to improve the quality of the article.

(1) We replaced reference [8] to express the latest research progress in the field of target detection in UAV aerial images. See line39 and line905-906.

(2) We replaced reference [17] to demonstrate the latest relevant research on autonomous driving technology. See line 135 and line 923.

(3) We added reference [33] in section 2.2 to provide the latest research background for aviation small target detection technology and demonstrate the importance of conducting research on aviation image small targets. See lines 217-222 and lines 951-952.

(4) We added reference [44] in section 2.3 to expand the research perspective of the article in the field of transformers and demonstrate new applications of transformers in image detection. See lines 252-260 and lines 943-944.

Point 2: The way data is presented in the manuscript could be improved. This includes clearer figures, tables, and possibly the addition of supplementary material to aid in the understanding of the research findings.

Response 2: We have carefully examined the relevant sections and supplemented Table 6 to help readers understand targets at different scales. See Line754-756 and Table 6.

We have zoomed in on Figure 14 (a-d) to present the detection results of small targets more clearly, making it easier to show the detection differences of the model.

We arranged the presentation of other data according to the presentation methods in similar research papers, such as:

â‘ Wan, Y.; Zhong, Y.; Huang, Y.; Han, Y.; Cui, Y.; Yang, Q.; Li, Z.; Yuan, Z.; Li, Q. ARSD: An Adaptive Region Selection Object Detection Framework for UAV Images. Drones 2022, 6, 228. https://doi.org/10.3390/drones6090228

②Hussain, R.; Karbhari, Y.; Ijaz, M.F.; Woźniak, M.; Singh, P.K.; Sarkar, R. Revise-Net: Exploiting Reverse Attention Mechanism for Salient Object Detection. Remote Sens. 2021, 13, 4941. https://doi.org/10.3390/rs13234941

â‘¢Lu, G.; He, X.; Wang, Q.; Shao, F.; Wang, H.; Wang, J. A Novel Multi-Scale Transformer for Object Detection in Aerial Scenes. Drones 2022, 6, 188. https://doi.org/10.3390/drones6080188

Point 3: The results and discussion sections could be made more comprehensive. Providing a deeper analysis of the findings and discussing their implications in a broader context would add value to the manuscript.

Response 3: Based on your suggestion, we have added some article contributions and future research content in the summary section to enhance the value of the article. See lines 866-870 and 877-879.

Point 4: Certain technical aspects, such as the accuracy of equations, the precision of technical terms, and the consistency of technical descriptions throughout the manuscript, could be revised for clarity and correctness.

Response 4: We have carefully examined the manuscript and revised it according to the aspects you put forward. See equations (1), (2), (3), line396-397, line405 and line634 for details We also adjusted the order of equations (17), (18) and (19).

Reviewer 2 Report

Comments and Suggestions for Authors

Review

 In the article, the authors proposed an improved YOLOv7 model that allows the detection of objects in aerial images or from UAVs of various sizes. Particular attention was paid to solving the problem of large-scale facilities. The authors presented in detail the differences between the classic YOLOv7 method and the proposed ATS-YOLOv7 method in a diagram.

The positives include a lot of literature from the last few years, as well as a comparison of the proposed method with State-of-the-Art deep learning models.

However, the article requires a few corrections and clarifications, especially related to the presented results.

1.      The abstract states that the object detection accuracy was 87% - does this apply to the detection of large, small or medium-sized objects? Is it the average of the three – maybe it needs to be separated. In the summary, also write what categories of objects were used for tests - list, for example, 2 from each group..

2.     The authors write that ATS-YOLOv7 additionally recognizes large objects 160x160 in comparison to YOLOv7 (Fig. 2 and 3). Figure 14a shows no difference in object detection using YOLOv7 and ATS-YOLOv7.

3.      In Table 6, it would be a good idea to mark small, medium and large objects and calculate the average mAP for such groups. Then it's easier to notice the differences.

4.      Fig. 14a shows that YOLOv5, YOLOv7 and ATS-YOLOv7 produce the same results. So what is the improvement in object detection in ATS-YOLOv7? Maybe it would be good to mark the differences in these images, because they are difficult to notice. Similarly for 14b, c and d.

5.      In Fig. 15, each image has different types of objects ?(Multi-scale, occluded, densely arranged and Tiny objects). What results were obtained using the ATS-YOLOv7 model if all four types of objects are present in the image at the same time?

Author Response

Response to Reviewer 2 Comments

Point 1: The abstract states that the object detection accuracy was 87% - does this apply to the detection of large, small or medium-sized objects? Is it the average of the three – maybe it needs to be separated. In the summary, also write what categories of objects were used for tests - list, for example, 2 from each group.

Response 1: (1) The object detection rate of 87% mentioned in the abstract refers to the mean average precision mAP of the model for all categories of objects (large, medium, and small) in the dataset. mAP is one of the most important standards for measuring the comprehensive detection performance of the model.

(2) This paper mainly detects and verifies the four kinds of objects: large, medium, small, and tiny. The last part of the summary summarizes the overall detection results of the four groups of objects, so as to reflect the overall performance of the proposed method. Therefore, it is not appropriate to show the test results of each group of single objectives here.

Point 2: The authors write that ATS-YOLOv7 additionally recognizes large objects 160x160 in comparison to YOLOv7 (Fig. 2 and 3). Figure 14a shows no difference in object detection using YOLOv7 and ATS-YOLOv7.

Response 2: (1) 160x160 here refers to the size of the feature map. In the feature mapping process of subsequent target prediction, one pixel in the feature image can cover four pixels of the original image, which makes the model able to detect tiny targets with fewer pixels (smaller size). See lines 322-331 for this explanation.

(2) Figure 14a shows the detection results of various models for multi-scale targets. The difference between the detection results of YOLOv7 and ATS-YOLOv7 is mainly in the small target part. Because the target scale is too small to be found, which is our reason, we processed the image to show the difference of detection results more clearly.

Point 3: In Table 6, it would be a good idea to mark small, medium and large objects and calculate the average mAP for such groups. Then it's easier to notice the differences.

Response 3: (1) We have improved table 6. We marked the scale type for each target. See line755-756 and table 6.

(2) mAP refers to the average value of AP (average precision) obtained by a model after detecting all types of targets in the dataset. It is an average value in itself, without calculating its average value. Here, each model corresponds to a mAP. Comparing the map values of each model to measure their detection ability is a common and important means in target detection experiments. Such as the article:

Lu, G.; He, X.; Wang, Q.; Shao, F.; Wang, H.; Wang, J. A Novel Multi-Scale Transformer for Object Detection in Aerial Scenes. Drones 2022, 6, 188. https://doi.org/10.3390/drones6080188

Point 4: Fig. 14a shows that YOLOv5, YOLOv7 and ATS-YOLOv7 produce the same results. So what is the improvement in object detection in ATS-YOLOv7? Maybe it would be good to mark the differences in these images, because they are difficult to notice. Similarly for 14b, c and d.

Response 4: Figure 14a shows that the detection results of YOLOv5, YOLOv7, and ATS-YOLOv7 are not the same, and their main difference lies in their ability to detect tiny objects. In question two, we adjusted all the images in Figure 14 to clearly show the detection differences of each model.

Point 5: In Fig. 15, each image has different types of objects? (Multi-scale, occluded, densely arranged and Tiny objects). What results were obtained using the ATS-YOLOv7 model if all four types of objects are present in the image at the same time?

Response 5: Overall, Figure 15 corresponds to the detection results of four types of targets: multi-scale, occlusion, dense arrangement, and tiny objects. It can be observed that the first multi-scale object detection image also contains car targets with occlusion, dense arrangement, and tiny scale (in the upper left corner of the image), all of which have been detected by ATS-YOLOv7.

Our detection objects come from image databases, and these images come from real scenes, which are not and are not easy to design artificially because they are large scenes. Therefore, we can only test the algorithm by selecting difficult samples from the database, and cannot design an ideal difficult sample.

Reviewer 3 Report

Comments and Suggestions for Authors

The research topic is interesting with a real engineering solution for UAV Aerial Images. The improved model is well presented. Experiments are well conducted with ablation texts.

The manuscript is suitable for Electronics after revising according to the following comments:

1)      When mentioning the IoU, SIoU, and CIoU, it is better to present the full definitions rather than just the abbreviations.

2)      The reference for ChatGPT is not adequate. This paper of 19 is not the original research of ChatGPT.

3)      The logical order to describe the related work needs to be adjusted. It is better to put the 2.2 YOLO as 2.1. And to change the 2.1 Aerial object detection to  2.2.

4)      As a part of the review of the history of YOLO from the first YOLO paper by Joseph Redmon [26] in 2016, it is better to mention the first research related to “Once Learning” by Weigang and Silva (1999).

5)      The reference in line 180 should be [26] not [25]. If you talk YOLO, it needs to cite the original paper [26].

6)      It is still not clear of SIoU in 2.4 Loss function and 3.1 Overview ….

7)      How about YOLOv6? As you mentioned that YOLOv7 was developed by [27], ATS-YOLOv7 should be proposed by your research according to 3.1 Overview … and also Figure 1 which there is no any reference in this figure.

8)      There is no any reference to Figure 2, it should be [27]. Actually, it is not necessary to put this figure because you need to get the authority of the original authors ([27] ?).

9)      In Line 366, please check: F6 is P5

10)   There are no references for all the figures, this means that all of them are the original research of this paper. Please check it.

11)   It is better to re-produce all equations such as using LaTex.

12)   Please check equations (16 - 17).

13)   It is better to use a table to present the cases described in 4.4.1 for more clarity.

14)   It is better to redesign some tables with the margin.

15)    The conclusion section needs to add some discussions of the contributions and future work to be more rich.

Ref. Weigang, Li, and Nilton Correia da Silva. "A study of parallel neural networks." IJCNN'99. International Joint Conference on Neural Networks. Proceedings (Cat. No. 99CH36339). Vol. 2. IEEE, 1999.

Comments on the Quality of English Language

English presentation is good, minor revising is necessary.

Author Response

Response to Reviewer 3 Comments

Point 1: When mentioning the IoU, SIoU, and CIoU, it is better to present the full definitions rather than just the abbreviations.

Response 1: We have added complete definitions of IoU, SIoU, and CIoU where they first appear in the text. See line 20, line 23, and line 275.

Point 2: The reference for ChatGPT is not adequate. This paper of 19 is not the original research of ChatGPT.

Response 2: After careful examination, we have re cited the reference in this area.

Reference:Radford, A.; Narasimhan, K. “Improving Language Understanding by Generative Pre-Training.” (2018). See line926.

Point 3: The logical order to describe the related work needs to be adjusted. It is better to put the 2.2 YOLO as 2.1. And to change the 2.1 Aerial object detection to 2.2.

Response 3: We have adjusted this part. See line146-222.

Point 4: As a part of the review of the history of YOLO from the first YOLO paper by Joseph Redmon [26] in 2016, it is better to mention the first research related to “Once Learning” by Weigang and Silva (1999).

Response 4: After studying and researching your article, we insert it into Section 2.1 as an important supplement to this part. See line147-152, line927.

Point 5: The reference in line 180 should be [26] not [25]. If you talk YOLO, it needs to cite the original paper [26].

Response 5: We have corrected the references in the original line180 and line184. See line152 and line156 after revision, and the document number is [21].

Point 6: It is still not clear of SIoU in 2.4 Loss function and 3.1 Overview…

Response 6: We have annotated the definition of SIoU, see Line23, and see section 3.2.3 for the internal details and specific parameter calculation relationship of SIoU. SIoU threshold is set in Non-Maximum Suppression in 3.1 Overview.

Point 7: How about YOLOv6? As you mentioned that YOLOv7 was developed by [27], ATS-YOLOv7 should be proposed by your research according to 3.1 Overview … and also Figure 1 which there is no any reference in this figure.

Response 7: (1) The article did not elaborate on YOLOv6, mainly because it has a similar model architecture and performance to YOLOv5. It can be said that it is the transitional stage from YOLOv5 to YOLOv7. Therefore, in the relevant work section of the article, we only introduced the highly representative YOLOv5 and YOLOv7 in the YOLO family.

(2) We have carefully considered your suggestion and, based on the situation in this article, we have deleted the YOLOv7 section (original section 3.2).

Point 8: There is no any reference to Figure 2, it should be [27]. Actually, it is not necessary to put this figure because you need to get the authority of the original authors ([27]?).

Response 8: Following your suggestion and careful consideration, we have removed the section regarding YOLOv7, which is the same as question 7. Adjust the remaining image numbers accordingly.

Point 9: In Line 366, please check: F6 is P5.

Response 9: We have checked the description of line 366 in the original text, where F6 is a new feature layer generated by P5 through the AAM module. It is now specifically explained in the parentheses of line 361.

Point 10: There are no references for all the figures, this means that all of them are the original research of this paper. Please check it.

Response 10: We confirm that all the figures are the original research of this article.

Point 11: It is better to re-produce all equations such as using LaTex.

Response 11: Thank you for your suggestion. We have made slight adjustments to all equations. After confirmation, the style of our formula meets the requirements of this journal.

Point 12: Please check equations (16 - 17).

Response 12: We carefully checked equations (16-17) and adjusted the order of equations (17), (18), and (19) to facilitate reader understanding.

Point 13: It is better to use a table to present the cases described in 4.4.1 for more clarity.

Response 13: Section 4.4.1 is the precision-recall experiment section, which aims to compare and analyze the superior performance of the proposed method by analyzing the detection results of various improved methods on various aerial datasets. The P-R curve generated from the experimental results is a necessary means for observing and analyzing data, and it is mainly used to compare the trends of data changes. Therefore, the experiment aims to display the macroscopic information composed of all points on each curve, The value of a single point alone cannot reflect the performance of the algorithm, which is also a common method in similar research work, such as the following two articles:

â‘ Wan, Y.; Zhong, Y.; Huang, Y.; Han, Y.; Cui, Y.; Yang, Q.; Li, Z.; Yuan, Z.; Li, Q. ARSD: An Adaptive Region Selection Object Detection Framework for UAV Images. Drones 2022, 6, 228. https://doi.org/10.3390/drones6090228

②Hussain, R.; Karbhari, Y.; Ijaz, M.F.; Woźniak, M.; Singh, P.K.; Sarkar, R. Revise-Net: Exploiting Reverse Attention Mechanism for Salient Object Detection. Remote Sens. 2021, 13, 4941. https://doi.org/10.3390/rs13234941

Point 14: It is better to redesign some tables with the margin.

Response 14: We have adjusted the margin of the table in this article.

Point 15: The conclusion section needs to add some discussions of the contributions and future work to be more rich.

Response 15: In the summary part, we added some contributions and future research work. See line866-870 and line877-879.

Round 2

Reviewer 2 Report

Comments and Suggestions for Authors

The authors comprehensively responded to my comments and made some changes that I asked for.

I recommend accepting this revised manuscript for publication.